# ADVANCING GRAPH GENERATION THROUGH BETA DIFFUSION

**Xinyang Liu**[1,*], **Yilin He**[1,*], **Bo Chen**[2], **Mingyuan Zhou**[1]
[1]The University of Texas at Austin, [2]Xidian University
xinyangatk@gmail.com, yilin.he@mccombs.utexas.edu
bchen@mail.xidian.edu.cn, mingyuan.zhou@mccombs.utexas.edu

## ABSTRACT

Diffusion models have excelled in generating natural images and are now being adapted to a variety of data types, including graphs. However, conventional models often rely on Gaussian or categorical diffusion processes, which can struggle to accommodate the mixed discrete and continuous components characteristic of graph data. Graphs typically feature discrete structures and continuous node attributes that often exhibit rich statistical patterns, including sparsity, bounded ranges, skewed distributions, and long-tailed behavior. To address these challenges, we introduce Graph Beta Diffusion (GBD), a generative model specifically designed to handle the diverse nature of graph data. GBD leverages a beta diffusion process, effectively modeling both continuous and discrete elements. Additionally, we propose a modulation technique that enhances the realism of generated graphs by stabilizing critical graph topology while maintaining flexibility for other components. GBD competes strongly with existing models across multiple general and biochemical graph benchmarks, showcasing its ability to capture the intricate balance between discrete and continuous features inherent in real-world graph data. Our PyTorch code is available at https://github.com/xinyangATK/GraphBetaDiffusion.

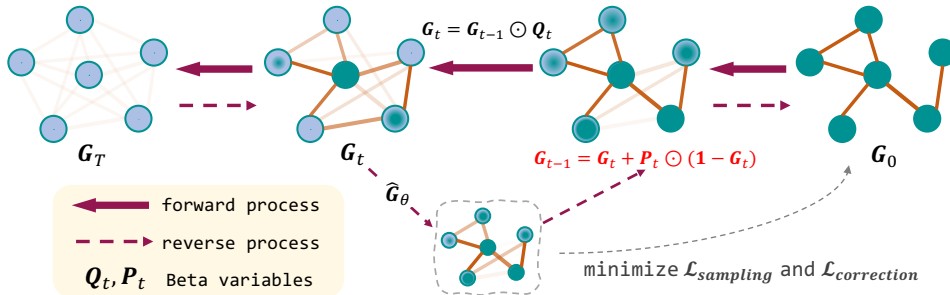

Figure 1: Overview of the forward and reverse diffusion processes of GBD. The multiplicative factors $\mathbf{Q}_t$ and $\mathbf{P}_t$ are sampled from beta distributions parameterized by the initial graphs $\mathbf{G}_0$ and "clean graphs" predicted by $\hat{G}_\theta$. The neural network $\hat{G}_\theta$ is learned through minimizing Equation 8 constituted by $\mathcal{L}_{\text{sampling}}$ and $\mathcal{L}_{\text{correction}}$.

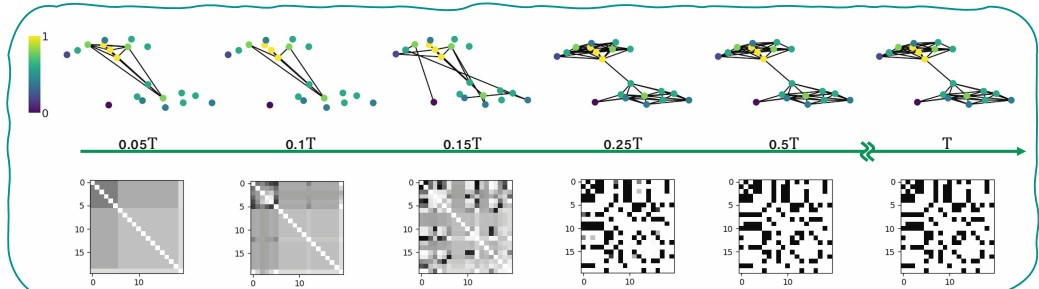

Figure 2: Edge generation process of the GBD model in graph topology (top) and adjacency matrix (bottom) views. Nodes are sorted by descending degree centrality and color-coded by degree, from yellow (high) to purple (low). The modulation technique in Section 2.3 enables early emergence of key positions such as community hubs, enhancing reverse chain stability.

---

[*]Equal contribution.

# 1 INTRODUCTION

In recent years, there has been a significant surge in interest and activity in the field of graph generation, particularly with the development of advanced generative models tailored for graphs. This growing attention is driven by the recognition of graph data's pervasive presence and utility across diverse real-world applications, ranging from social network study (Newman et al., 2002; Conti et al., 2011; Abbe, 2018; Shirzadian et al., 2023) to biochemical molecular research (Jin et al., 2018; Liu et al., 2018; Bongini et al., 2021; Guo et al., 2024; Li & Yamanishi, 2024). Additionally, the rapid evolution of machine learning tools has introduced powerful techniques for data generation, among which diffusion models (Ho et al., 2020; Song et al., 2021; Austin et al., 2021; Avdeyev et al., 2023; Chen & Zhou, 2023; Zhou et al., 2023) stand out as a notable example. As these advanced tools intersect with the task of graph generation, we witness the emergence of numerous diffusion-based graph generative models (Niu et al., 2020a; Jo et al., 2022; Haefeli et al., 2022; Huang et al., 2022; Vignac et al., 2023; Jo et al., 2023; Cho et al., 2023; Chen et al., 2023; Kong et al., 2023).

While diffusion-based graph generative models often demonstrate superior performance compared to their predecessors (You et al., 2018; De Cao & Kipf, 2018; Li et al., 2018; Simonovsky & Komodakis, 2018; Liu et al., 2019; Shi et al., 2020; Luo et al., 2021; Martinkus et al., 2022), there is still potential for further enhancement in the quality of generated graphs. Among the latest advancements in these methods, it is widely recognized that incorporating inductive bias from the graph data is generally beneficial for model design (Jo et al., 2023). One promising direction of incorporating this bias involves considering the statistical characteristics of the distribution of graph data. For instance, both Graph D3PM (Haefeli et al., 2022) and DiGress (Vignac et al., 2023) have demonstrated that when considering the binary or categorical nature of the graph adjacency matrix and modeling it in the discrete space, it provides benefits for generating more realistic graphs.

Accounting for the discreteness of the graph adjacency matrix has shown enhancement to performance. However, it is crucial to recognize that the complexity and flexibility of the distribution characteristics of graph data extend beyond mere discreteness. Real-world graphs usually display sparse edge distributions and exhibit diverse statistical patterns in node attributes, which may include skewed, multi-modal, or long-tailed distributions (Barabási et al., 2000; Ciotti et al., 2015; Liang et al., 2023; Wang et al., 2023). While the values within node feature matrices may not inherently bounded by range, they are often be empirically represented or processed into quantities that are bound by specific limits. Considering the unique characteristics inherent to graph data, it is clear that Gaussian and categorical distributions, often default choices for constructing diffusion processes, may not adequately align with these graph traits. This misalignment could introduce noticeable limitations in accurately modeling the distribution of graphs.

Given the unique statistical characteristics of graph data, the beta distribution emerges as a particularly suitable modeling choice. With great flexibility to model continuous data with various statistical characteristics and approximate discrete distributions at all sparsity levels, the beta distribution aligns well with the inherent traits of graphs, hence making itself a promising candidate to surpass the potential limitations imposed by utilizing Gaussian or categorical distributions. In this paper, we introduce Graph Beta Diffusion (GBD) as a novel addition to diffusion-based graph generative models. GBD models the joint distribution of node attributes and edge connections within a graph through beta diffusion (Zhou et al., 2023), a generative diffusion process that is built upon the thinning of beta random variables in its multiplicative forward diffusion process and the thickening in its multiplicative reverse process.

We underscore two major contributions arising from the development of GBD. First, our experiments generating data on various synthetic and real-world graphs confirm the effectiveness of beta diffusion as a strategic choice within the design framework of the backbone diffusion model, especially for graph generation tasks. Second, our exploration of the model's design space has yielded a set of recommended practices, notably a novel modulation technique that bolsters the stability of generating essential graph structures. We demonstrate that these practices, when implemented together, lead to consistent enhancements in model performance.

# 2 THE METHODOLOGY

In this study, our primary focus lies in generating two types of graphs: generic graphs and molecular graphs. A graph with $N$ nodes is represented by the tuple $\mathbf{G} = (\mathbf{A}, \mathbf{X})$, where $\mathbf{X} \in \mathbb{R}^{N \times D}$

denotes the node features with feature dimension $D$, and $\mathbf{A} \in \mathbb{R}^{N \times N}$ is the symmetric binary adjacency matrix that defines the connections between nodes. The selection of node features offers high flexibility, ranging from raw-data-provided node categories to hand-crafted features such as node-level statistics (Jo et al., 2022) or spectral graph signals (Jo et al., 2023). The features within $\mathbf{X}$ exhibit great diversity in their nature, including numerical, categorical, and ordinal types. Through preprocessing methods including dummy-encoding, empirical CDF transformation or normalization, we standardize them as continuous variables bounded by $[0, 1]$. For molecular graphs, we use $\mathbf{A}^{(1:K)}$ to represent the structure of a graph with $K$ types of edges and $\mathbf{G}$ is defined as $(\mathbf{A}^{(1:K)}, \mathbf{X})$. In the sequel, we by default employ the generic graph scenario to illustrate the methodology.

## 2.1 Forward and reverse beta diffusion processes

**Forward multiplicative beta diffusion process.** Such a process can be characterized by the transition probability $q(\mathbf{G}_t \mid \mathbf{G}_{t-1}, \mathbf{G}_0)$, with $\mathbf{G}_0$ denoting the combination of the original adjacency matrix and node feature matrix. Following recent diffusion-based graph generative models (Jo et al., 2022; Vignac et al., 2023; Jo et al., 2023; Cho et al., 2023), we assume $q(\mathbf{G}_t \mid \mathbf{G}_{t-1}, \mathbf{G}_0)$ to be factorizable such that $q(\mathbf{G}_t \mid \mathbf{G}_{t-1}, \mathbf{G}_0) = q(\mathbf{A}_t \mid \mathbf{A}_{t-1}, \mathbf{A}_0) \cdot q(\mathbf{X}_t \mid \mathbf{X}_{t-1}, \mathbf{X}_0)$. Constructing the forward multiplicative beta diffusion process (Zhou et al., 2023) for graph modeling, we have:

$$\mathbf{A}_t = \mathbf{A}_{t-1} \odot \mathbf{Q}_{A,t}, \ \mathbf{Q}_{A,t} \sim \text{Beta}\left(\eta_A \alpha_t \mathbf{A}_0, \eta_A(\alpha_{t-1} - \alpha_t)\mathbf{A}_0\right), \tag{1}$$

$$\mathbf{X}_t = \mathbf{X}_{t-1} \odot \mathbf{Q}_{X,t}, \ \mathbf{Q}_{X,t} \sim \text{Beta}\left(\eta_X \alpha_t \mathbf{X}_0, \eta_X(\alpha_{t-1} - \alpha_t)\mathbf{X}_0\right), \ t \in [1, T]. \tag{2}$$

Here $\eta_A, \eta_X$ are positive scalars adjusting the concentration of beta distributions, with higher values leading to enhanced concentration and reduced variability. The diffusion noise schedule is defined with $\{\alpha_t \mid t \in [1, T]\}$, which represent a sequence of values descending from 1 towards 0 as $t$ increases. Elements in the fractional multiplier $\mathbf{Q}_{A,t}$ or $\mathbf{Q}_{A,t}$ are independently sampled from their respective beta distributions. With the forward diffusion process defined in Equations 1 and 2, we characterize the stochastic transitions of an element $g$ within $\mathbf{G}$ as:

$$q(g_t \mid g_{t-1}, g_0) = \frac{1}{g_{t-1}} \text{Beta}\left(\frac{g_t}{g_{t-1}} \mid \eta \alpha_t g_0, \eta(\alpha_{t-1} - \alpha_t)g_0\right), \tag{3}$$

where depending on whether $g$ is an element in $\mathbf{A}$ or $\mathbf{X}$, we have either $\eta = \eta_A$ or $\eta = \eta_X$. Derived from Equation 3, the joint distribution $q(\mathbf{G}_{1:T} \mid \mathbf{G}_0)$ has analytical format in the marginal distribution at each time stamp $t$, specifically,

$$q(\mathbf{G}_t \mid \mathbf{G}_0) = \text{Beta}(\eta \alpha_t \mathbf{G}_0, \eta(1 - \alpha_t \mathbf{G}_0)). \tag{4}$$

**Reverse multiplicative beta diffusion process.** It is important to note that the joint distribution $q(\mathbf{G}_{1:T} \mid \mathbf{G}_0)$ can be equivalently constructed in reverse order, which directs samples from the terminus states $\mathbf{G}_T$ towards the initial states $\mathbf{G}_0$ by incrementally applying the changes $\delta \mathbf{G}_t$ at each reversed time stamp. With the changes at a given time $t$ parameterized as $\delta \mathbf{G}_t := \mathbf{P}_t \odot (1 - \mathbf{G}_t)$, where $\mathbf{P}_t$ are beta-distributed fractional multipliers, the time-reversal multiplicative sampling process can be mathematically defined as: for $t = T, T-1, \cdots, 1$,

$$\mathbf{G}_{t-1} = \mathbf{G}_t + \mathbf{P}_t \odot (1 - \mathbf{G}_t), \ \mathbf{P}_t \sim \text{Beta}\left(\eta(\alpha_{t-1} - \alpha_t)\mathbf{G}_0, \eta(1 - \alpha_{t-1}\mathbf{G}_0)\right). \tag{5}$$

Similar to the forward sampling process, we can derive the transition distribution corresponding to the reverse sampling process described in Equation 5 as following:

$$q(\mathbf{G}_{t-1} \mid \mathbf{G}_t, \mathbf{G}_0) = \frac{1}{1 - \mathbf{G}_t} \text{Beta}\left(\frac{\mathbf{G}_{t-1} - \mathbf{G}_t}{1 - \mathbf{G}_t} \mid \eta(\alpha_{t-1} - \alpha_t)\mathbf{G}_0, \eta(1 - \alpha_{t-1}\mathbf{G}_0)\right). \tag{6}$$

Following previous work (Austin et al., 2021; Haefeli et al., 2022; Vignac et al., 2023; Zhou et al., 2023), we construct the reverse diffusion process through the definition of ancestral sampling distribution as following:

$$p_\theta(\mathbf{G}_{t-1} \mid \mathbf{G}_t) := q(\mathbf{G}_{t-1} \mid \mathbf{G}_t, \hat{G}_\theta(\mathbf{G}_t, t)), \tag{7}$$

where $\hat{G}_\theta(\mathbf{G}_t, t)$ is a neural network that predicts the conditional expectation of $\mathbf{G}_0$ given $\mathbf{G}_t$. Following Vignac et al. (2023), we instantiate $\hat{G}_\theta(\mathbf{G}_t, t)$ as a graph transformer network (Dwivedi & Bresson, 2020). We present the complete sampling process in Appendix D.1.

## 2.2 Training GBD

The overall training procedure of GBD is described in Section D.1 of the Appendix. We employ the objective function proposed by beta diffusion (Zhou et al., 2023), specifically,

$$\mathcal{L} = \sum_{t=2}^{T} (1-\omega)\mathcal{L}_{\text{sampling}}(t, \mathbf{G}_0) + \omega\, \mathcal{L}_{\text{correction}}(t, \mathbf{G}_t),\ \omega \in [0, 1]. \tag{8}$$

In Equation 8, the loss terms associated with sampling and correction are defined as

$$\mathcal{L}_{\text{sampling}}(t, \mathbf{G}_0) \triangleq \mathbb{E}_{q(\mathbf{G}_t, \mathbf{G}_0)} \text{KL}\left(p_\theta(\mathbf{G}_{t-1} \mid \mathbf{G}_t) \parallel q(\mathbf{G}_{t-1} \mid \mathbf{G}_t, \mathbf{G}_0)\right), \tag{9}$$

$$\mathcal{L}_{\text{correction}}(t, \mathbf{G}_0) \triangleq \mathbb{E}_{q(\mathbf{G}_t, \mathbf{G}_0)} \text{KL}\left(q(\mathbf{G}_\tau \mid \hat{G}_\theta(\mathbf{G}_t, t)) \parallel q(\mathbf{G}_\tau \mid \mathbf{G}_0)\right). \tag{10}$$

In Equation 10, the KL divergence is evaluated between the following distributions: $q(\mathbf{G}_\tau \mid \hat{G}_\theta(\mathbf{G}_t, t))$ is $\text{Beta}(\eta\alpha_t\hat{G}_\theta(\mathbf{G}_t, t), \eta(1 - \alpha_t\hat{G}_\theta(\mathbf{G}_t, t)))$, and $q(\mathbf{G}_\tau \mid \mathbf{G}_0)$ is the same as $q(\mathbf{G}_t \mid \mathbf{G}_0)$ in distribution. The subscript $\tau$ is introduced to represent a generic graph sample other than $\mathbf{G}_t$ that is also obtained at time $t$ from the forward diffusion process. The core principle behind the loss function terms can be described as follows: $\mathcal{L}_{\text{sampling}}$ drives the empirical ancestral sampling distribution towards the destination-conditional posterior distribution, while $\mathcal{L}_{\text{correction}}$ corrects the bias on marginal distribution at each time stamp accumulated through the ancestral sampling. These two types of loss terms collectively reduce the divergence between the empirical joint distribution on two graphs sampled from adjacent time stamps in the reverse process, and their joint distribution derived from the forward diffusion process. A positive weight $\omega$ is introduced to balance the effects of these two types of loss terms. We set it to 0.01, following Zhou et al. (2023), and found that this configuration is sufficient to produce graphs that closely resemble the reference graphs without further tuning. To better elucidate the optimization objective, we list out the analytical expressions of the KL divergence term in Appendix B.

It is demonstrated in Zhou et al. (2023) that the KL divergence between two beta distributions can be expressed in the format of a Bregman divergence. Namely, considering a convex function $\phi(\alpha, \beta) \triangleq \ln \text{Beta}(\alpha, \beta)$, where $\text{Beta}(\alpha, \beta) = \frac{\Gamma(\alpha)\Gamma(\beta)}{\Gamma(\alpha+\beta)}$ is the beta function, the loss term $\mathcal{L}_{\text{sampling}}$ can be expressed as

$$\mathcal{L}_{\text{sampling}}(t, \mathbf{G}_0) = \mathbb{E}_{q(\mathbf{G}_t)}\mathbb{E}_{q(\mathbf{G}_0 \mid \mathbf{G}_t)}d_\phi\left([\mathbf{a}_{\text{sampling}}, \mathbf{b}_{\text{sampling}}], [\mathbf{a}_{\text{sampling}}^*, \mathbf{b}_{\text{sampling}}^*]\right),$$
$$\mathbf{a}_{\text{sampling}} = \eta(\alpha_{t-1} - \alpha_t)\mathbf{G}_0,\ \mathbf{b}_{\text{sampling}} = \eta(1 - \alpha_{t-1}\mathbf{G}_0), \tag{11}$$
$$\mathbf{a}_{\text{sampling}}^* = \eta(\alpha_{t-1} - \alpha_t)\hat{\mathbf{G}}_0,\ \mathbf{b}_{\text{sampling}}^* = \eta(1 - \alpha_{t-1}\hat{\mathbf{G}}_0).$$

Likewise, we can express the correction loss term $\mathcal{L}_{\text{correction}}$ as

$$\mathcal{L}_{\text{correction}}(t, \mathbf{G}_0) = \mathbb{E}_{q(\mathbf{G}_t)}\mathbb{E}_{q(\mathbf{G}_0 \mid \mathbf{G}_t)}d_\phi\left([\mathbf{a}_{\text{correction}}, \mathbf{b}_{\text{correction}}], [\mathbf{a}_{\text{correction}}^*, \mathbf{b}_{\text{correction}}^*]\right),$$
$$\mathbf{a}_{\text{correction}} = \eta\alpha_t\mathbf{G}_0,\ \mathbf{b}_{\text{correction}} = \eta(1 - \alpha_t\mathbf{G}_0), \tag{12}$$
$$\mathbf{a}_{\text{correction}}^* = \eta\alpha_t\hat{\mathbf{G}}_0,\ \mathbf{b}_{\text{correction}}^* = \eta(1 - \alpha_t\hat{\mathbf{G}}_0).$$

Here we reference the $d_\phi$ notation of Banerjee et al. (2005) to represent the Bregman divergence. As established in Lemmas 3–5 of Zhou et al. (2023), since both $\mathcal{L}_{\text{sampling}}$ and $\mathcal{L}_{\text{correction}}$ can be formulated as Bregman divergences, Proposition 1 of Banerjee et al. (2005) applies, demonstrating that they yield the same unbiased optimal solution. This justifies the use of $\hat{\mathbf{G}}_0$ in the reverse diffusion process, as detailed below.

**Property 1** *Both $\mathcal{L}_{\text{sampling}}$ and $\mathcal{L}_{\text{correction}}$ are uniquely minimized at*

$$\hat{\mathbf{G}}_0 = \hat{G}_\theta(\mathbf{G}_t, t) = \mathbb{E}_{q(\mathbf{G}_0 \mid \mathbf{G}_t)}[\mathbf{G}_0].$$

## 2.3 Exploring the design space of GBD

Many diffusion-based graph generative models offer great flexibility with technical adjustment to enhance their practical performances. Here we list four impactful dimensions among the design space of GBD. Namely, data transformation, concentration modulation, logit-domain computation, and neural-network precondition. We elaborate each design dimension below and discuss our choices in these aspects in the Appendix.

**Data transformation.** We convert the raw data $(\mathbf{A}, \mathbf{X})$ to $\mathbf{G}_0$ through linear transformations, *i.e.*,

$$\mathbf{G}_0 = (\mathbf{A}_0, \mathbf{X}_0), \text{ where } \mathbf{A}_0 = w_A \cdot \mathbf{A} + b_A, \ \mathbf{X}_0 = w_X \cdot \mathbf{X} + b_X, \quad (13)$$

with the constraints that $\min(w_A, b_A, w_X, b_X) > 0$ and $\max(w_A + b_A, w_X + b_X) \leq 1$. This operation not only ensure that all data values fall within the positive support of beta distributions, avoiding gradient explosion when optimizing the loss function, but also provide an effective means to adjust the rate at which diffusion trajectories mix. A forward diffusion trajectory reaches a state of "mix" when it becomes indistinguishable to discern the initial value from its counterfactual given the current value. A suitable mixing rate ensures that the signal-to-noise ratio (SNR) of the final state in the forward diffusion process approaches zero, meeting the prerequisite for learning reverse diffusion while preserving the learnability of graph structural patterns. The scaling parameter provides a macro control for the mixing rate, with a smaller value contracting the data range and promoting the arrival of the mixing state.

**Concentration modulation.** Another hyperparameter that offers a more refined adjustment to the mixing rate is the concentration parameter $\eta$. Higher values of $\eta$ reduce the variance of the fractional multipliers $\mathbf{P}_t$ sampled from their corresponding beta distributions, thus delaying the arrival of the mixing state. Leveraging this property, we have devised a simple yet effective modulation strategy to differentiate the mixing times across various graph substructures.

Specifically, we assign higher $\eta$ values to "important positions" within a graph, such as edges connecting high-centrality nodes or edges deemed significant based on domain knowledge, such as the carbon-carbon bond in chemical molecules. For instance, when modulating $\eta$ from degree centrality, the exact operation executed upon the $\eta$ values for edge $(u, v)$ and for the features of node $u$ can be mathematically expressed as

$$\eta_{u,v} = g_A(\max(\deg(u), \deg(v))), \quad \eta_u = g_X(\deg(u)). \quad (14)$$

Here we first prepare several levels of $\eta$ values, then utilize two assignment functions, namely $g_A(\cdot)$ and $g_X(\cdot)$, to map the node degrees (or their percentile in the degree population within one graph) to one of the choices of the $\eta$ values. We have observed that this operation indeed prolongs the presence of these substructures during the forward diffusion process, which in turn leads to their earlier emergence compared to the rest of the graph during the reverse process. Additionally, we provide an alternate definition of "importance positions" using betweenness centrality (Freeman, 1977), detailed in Appendix D.2, and also ablate its effects in Section 4.3.

We visualize the reverse process from two perspectives in Figure 2. We first obtain the $\eta_{u,v}$ by degrees retrieved from the training set before sampling and then generate graph through reverse beta diffusion. From the top row, we observe that edges linked to nodes with higher degrees (indicated by brighter colors) appear first, followed by other edges. From the bottom row, it is evident that edges connected to the first five nodes, which have higher degrees, are identified first and then progressively in descending order of degree. Notably, the nodes of the adjacency matrices in the bottom row are reordered by decreasing node degree of the final graph. Additionally, we can also find the predicted graph of GBD converges in an early stage to the correct topology. We attribute the enhanced quality of generated graphs to the early emergence of these "important substructures," which likely improves the reliability of generating realistic graph structures. Furthermore, this approach is particularly appealing as it allows for the flexible integration of graph inductive biases within the diffusion model framework.

**Logit domain computation.** Another noteworthy designing direction lies in the computation domain. Although the reverse sampling process directly implemented from Equation 5 is already effective to generate realistic graph data, we observe that migrating the computation to the logit space further enhances model performance and accelerates training convergence. One potential explanation is that the logit transformation amplifies the structural patterns of the graph when all edge weights are very close to zero at the beginning of the ancestral sampling process. Equivalent to Equation 5, the logit-domain computation can be expressed as

$$\text{logit}(\mathbf{G}_{t-1}) = \ln\left(e^{\text{logit}(\mathbf{G}_t)} + e^{\text{logit}(\mathbf{P}_t)} + e^{\text{logit}(\mathbf{G}_t)+\text{logit}(\mathbf{P}_t)}\right). \quad (15)$$

**Neural-network precondition.** Finally, we employ the neural-network precondition technique (Karras et al., 2022) and customize it for training GBD, which involves standardizing $\mathbf{G}_t$ before

passing them to the prediction network $\hat{G}_\theta(\cdot)$. In other words, we modify Equation 7 as

$$p_\theta(\mathbf{G}_{t-1} \mid \mathbf{G}_t) := q(\mathbf{G}_{t-1} \mid \mathbf{G}_t, \hat{G}_\theta(\tilde{\mathbf{G}}_t, t)), \quad \tilde{\mathbf{G}}_t = \frac{\mathbf{G}_t - \mathbb{E}[g_t]}{\sqrt{\mathrm{Var}[g_t]}} \text{ or } \frac{\mathrm{logit}(\mathbf{G}_t) - \mathbb{E}[\mathrm{logit}(g_t)]}{\sqrt{\mathrm{Var}[\mathrm{logit}(g_t)]}}. \quad (16)$$

To illustrate the application of neural-network preconditioning, we present an example of training GBD to generate graphs $\mathbf{G} = (\mathbf{A}, \mathbf{X})$. For simplicity, we assume the predictor operates in the original domain, with corresponding results for cases involving logit domain computations provided in Appendix C. We make following statistical assumptions regarding the marginal distribution of $a_0$ and $x_0$, the elements within the graph adjacency matrices and node feature matrices after the data transformation step: $a_0$ follows a Bernoulli distribution with two possible outcomes $a_{\min}, a_{\max}$, where the probability of $a_0 = a_{\max}$ is $p$, and $x_0$ follows a uniform distribution over the support $[x_{\min}, x_{\max}]$. Given that $g_t \mid g_0 \sim \mathrm{Beta}(\eta \alpha_t g_0, \eta(1 - \alpha_t g_0))$, and by applying the law of total expectation and the law of total variance, one can derive that

$$\mathbb{E}[a_t] = \alpha_t \left( p \cdot a_{\max} + (1 - p) \cdot a_{\min} \right),$$

$$\mathrm{Var}[a_t] = \frac{1}{\eta_A + 1} \left( \mathbb{E}[a_t] - \mathbb{E}[a_t]^2 \right) + \frac{\eta_A}{\eta_A + 1} \left( \alpha_t^2 (p(1-p))(a_{\max} - a_{\min})^2 \right), \quad (17)$$

$$\mathbb{E}[x_t] = \frac{1}{2} \alpha_t (x_{\min} + x_{\max}),$$

$$\mathrm{Var}[x_t] = \frac{1}{\eta_X + 1} \left( \mathbb{E}[x_t] - \mathbb{E}[x_t]^2 \right) + \frac{\eta_X}{12(\eta_X + 1)} \left( \alpha_t^2 (x_{\max} - x_{\min})^2 \right). \quad (18)$$

Using these computed quantities, we complete the neural-network preconditioning by standardizing $\mathbf{G}_t$ into $\tilde{\mathbf{G}}_t = (\tilde{\mathbf{A}}_t, \tilde{\mathbf{X}}_t)$, where the adjacency matrix is transformed as $\tilde{\mathbf{A}}_t = (\mathbf{A}_t - \mathbb{E}[a_t])/\sqrt{\mathrm{Var}[a_t]}$ and the node feature matrix as $\tilde{\mathbf{X}}_t = (\mathbf{X}_t - \mathbb{E}[x_t])/\sqrt{\mathrm{Var}[x_t]}$. The standardized graph $\tilde{\mathbf{G}}_t$ is then used as input to the predictor network. The derivation of Equations 17 and 18 is in Appendix C. While neural-network preconditioning is generally effective for stabilizing model training, we empirically find it to be particularly beneficial when training the GBD with the predictor defined in the logit space, where increased variability is introduced.

## 3 RELATED WORK

**Graph generative models.** Early attempts at modeling graph distributions trace back to the Erdős–Rényi graph model (ERDdS & R&wi, 1959; Erdős et al., 1960), from which a plethora of graph generative models have emerged. These models employ diverse approaches to establish the data generative process and devise optimization objectives, which in turn have significantly expanded the flexibility in modeling the distribution of graph data. Stochastic blockmodels (Holland et al., 1983; Lee & Wilkinson, 2019), latent variable models (Airoldi et al., 2009; Zhou, 2015; Caron & Fox, 2017), and their variational-autoencoder-based successors (Kipf & Welling, 2016; Hasanzadeh et al., 2019; Mehta et al., 2019; He et al., 2022) assume that edges are formed through independent pairwise node interactions, and thus factorize the probability of the graph adjacency matrix into the dot product of factor representations of nodes. Sequential models (You et al., 2018; Wang et al., 2018; Jin et al., 2018; Han et al., 2023) adopt a similar concept of node interactions but correlate these interactions by organizing them into a series of connection events. Additionally, some models treat the graph adjacency matrix as a parameterized random matrix and generate it by mapping a random vector through a feed-forward neural network (Simonovsky & Komodakis, 2018; De Cao & Kipf, 2018). In terms of optimization targets, many utilize log-likelihood-based objectives such as negative log-likelihood (Liu et al., 2019) or evidence lower bound objectives (Kipf & Welling, 2016), while others employ generative adversarial losses (Wang et al., 2018) or reinforcement learning losses (De Cao & Kipf, 2018). Diffusion-based graph generative models (Jo et al., 2022; Haefeli et al., 2022; Vignac et al., 2023; Niu et al., 2020b;a; Chen et al., 2023; Cho et al., 2023; Kong et al., 2023), including this work, feature a unique data generation process compared to previous models. They map the observed graph structures and node features to a latent space through a stochastic diffusion process, whose reverse process can be learned by optimizing a variational lower bound (Vignac et al., 2023) or numerically solving a reverse stochastic differential equation (Jo et al., 2022).

**Diffusion models.** The stochastic diffusion process is introduced by Sohl-Dickstein et al. (2015) for deep unsupervised learning, and its foundational connection with deep generative models is laid

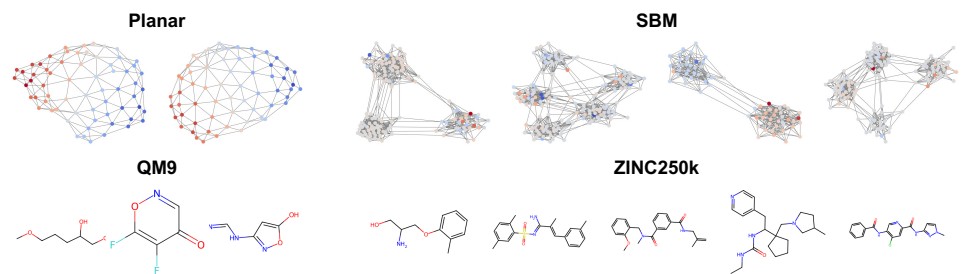

Figure 3: Examples of graphs generated by the GBD model on `Planar`, SBM, `QM9`, and `ZINC250k` datasets.

down by Song & Ermon (2019) and denoising diffusion probabilistic model (DDPM) (Ho et al., 2020). DDPM maps a data sample to the latent space via a Markov process that gradually applies noise to the original sample, and learns a reverse process to reproduce the sample in finite steps. The optimization and sampling processes in DDPM can be interpreted through the lens of variational inference (Sohl-Dickstein et al., 2015; Ho et al., 2020) or can be formulated as score matching with Langevin dynamics (Song et al., 2021; Song & Ermon, 2019). Both approaches are focused on diffusion processes that define the transition between normally distributed variables, which are proven effective for generating natural images. As the scope of generative tasks expands to discrete domains like text, diffusion models transitioning between discrete states emerge (Austin et al., 2021), which demonstrates that the choice of probabilistic distribution for modeling each noise state could significantly impact the learning task. This conclusion is also validated in the application of graph generation (Haefeli et al., 2022; Vignac et al., 2023). Further studies (Chen & Zhou, 2023; Zhou et al., 2023) are conducted to improve diffusion models by introducing novel diffusion processes based on probabilistic distributions that better capture the intrinsic characteristics of the generation target. Among these, the beta diffusion of Zhou et al. (2023) is chosen as the foundation of our method, due to the beta distribution's proficiency in capturing sparsity and modeling range-bounded data across mixed types. These traits are commonly observed in real-world graphs (Barabási et al., 2000; Ciotti et al., 2015; Liang et al., 2023; Wang et al., 2023).

## 4 EXPERIMENTS

### 4.1 GENERIC GRAPH GENERATION

**Datasets and metrics.** We use five graph datasets with varying sizes, connectivity, and topology, commonly employed as benchmarks. `Ego-small` includes 200 sub-graphs from the Citeseer network with $4 \leq N \leq 18$ nodes. `Community-small` has 100 synthetic graphs with $12 \leq N \leq 20$. `Grid` contains 100 2D grid graphs with $100 \leq N \leq 400$. `Planar` includes 200 synthetic planar graphs with $N = 64$. SBM comprises 200 stochastic block model graphs with 2–5 communities, where $44 \leq N \leq 187$ and each community has 20–40 nodes.

For a fair comparison, we follow the experimental setup of Jo et al. (2022; 2023), using the same train/test split. We evaluate using maximum mean discrepancy (MMD) (Gretton et al., 2012) to compare the distributions of key graph properties between test graphs and generated graphs: degree (**Deg.**), clustering coefficient (**Clus.**), and 4-node orbit counts (**Orbit**). For more complex structures like `Planar` and SBM, we also report the eigenvalues of the graph Laplacian (**Spec.**) and the percentage of valid, unique, and novel graphs (**V.U.N.**) to assess how well the model captures both intrinsic features and global graph properties. A lower score indicates better performance for all metrics except V.U.N. More details on these metrics can be found in Appendix E.1.

**Baselines.** We compare GBD against various graph generation methods, including autoregressive models: **DeepGMG** (Li et al., 2018), **GraphRNN** (You et al., 2018), and **GraphAF** (Shi et al., 2020); one-shot model: **GraphVAE** (Simonovsky & Komodakis, 2018); and flow-based models: **GNF** (Liu et al., 2019) and **GraphDF** (Luo et al., 2021). Additionally, we compare against state-of-the-art (SOTA) diffusion-based graph generative models, including score-based or continuous: **EDP-GNN** (Niu et al., 2020a), **GDSS**, **GDSS + Transofrmer (TF)** (Jo et al., 2022), **GruM** (Jo et al., 2023), and **ConGress** (Vignac et al., 2023); discrete: **DiGress** (Vignac et al., 2023); autoregressive: **GraphARM** (Kong et al., 2023); and a model with wavelet features: **Wave-GD** (Cho et al., 2023).

Table 1: MMD results for simple generic graph generation.

| Method | Ego-small | | | Community-small | | | Grid | | |
|---|---|---|---|---|---|---|---|---|---|
| | Deg. | Clus. | Orbit. | Deg. | Clus. | Orbit. | Deg. | Clus. | Orbit. |
| DeepGMG | 0.040 | 0.100 | 0.020 | 0.220 | 0.950 | 0.400 | - | - | - |
| GraphRNN | 0.090 | 0.220 | 0.003 | 0.080 | 0.120 | 0.040 | 0.064 | 0.043 | 0.021 |
| GraphAF | 0.030 | 0.110 | 0.001 | 0.180 | 0.200 | 0.020 | - | - | - |
| GraphDF | 0.040 | 0.130 | 0.010 | 0.060 | 0.120 | 0.030 | - | - | - |
| GraphVAE | 0.130 | 0.170 | 0.050 | 0.350 | 0.980 | 0.540 | 1.619 | **0.0** | 0.919 |
| GNF | 0.030 | 0.100 | 0.001 | 0.200 | 0.200 | 0.110 | - | - | - |
| EDP-GNN | 0.052 | 0.093 | 0.007 | 0.053 | 0.144 | 0.026 | 0.455 | 0.238 | 0.328 |
| GDSS | 0.021 | 0.024 | 0.007 | 0.045 | 0.086 | 0.007 | 0.111 | 0.005 | 0.070 |
| ConGress* | 0.037 | 0.064 | 0.017 | 0.020 | 0.076 | 0.006 | - | - | - |
| DiGress | 0.017 | 0.021 | 0.010 | 0.028 | 0.115 | 0.009 | - | - | - |
| GraphARM | 0.019 | 0.017 | 0.010 | 0.034 | 0.082 | 0.004 | - | - | - |
| Wave-GD | 0.012 | **0.010** | 0.005 | 0.007 | **0.058** | 0.002 | 0.144 | 0.004 | **0.021** |
| **GBD** | **0.011** ($\pm$0.008) | 0.014 ($\pm$0.013) | **0.002** ($\pm$0.004) | **0.002** ($\pm$0.003) | 0.060 ($\pm$0.004) | **0.002** ($\pm$0.003) | **0.045** ($\pm$0.004) | 0.011 ($\pm$0.002) | 0.040 ($\pm$0.014) |

Table 2: MMD results for complex generic graph generation.

| Method | Planar | | | | | SBM | | | | |
|---|---|---|---|---|---|---|---|---|---|---|
| | Deg. | Clus. | Orbit. | Spec. | V.U.N. | Deg. | Clus. | Orbit. | Spec. | V.U.N. |
| Training set | 0.0002 | 0.0310 | 0.0005 | 0.0052 | 100.0 | 0.0008 | 0.0332 | 0.0255 | 0.0063 | 100.0 |
| GraphRNN | 0.0049 | 0.2779 | 1.2543 | 0.0459 | 0.0 | 0.0055 | 0.0584 | 0.0785 | 0.0065 | 5.0 |
| GRAN | 0.0007 | 0.0426 | 0.0009 | 0.0075 | 0.0 | 0.0113 | 0.0553 | 0.0540 | 0.0054 | 25.0 |
| SPECTRE | 0.0005 | 0.0785 | 0.0012 | 0.0112 | 25.0 | 0.0015 | 0.0521 | **0.0412** | 0.0056 | 52.5 |
| EDP-GNN | 0.0044 | 0.3187 | 1.4986 | 0.0813 | 0.0 | 0.0011 | 0.0552 | 0.0520 | 0.0070 | 35.0 |
| GDSS | 0.0041 | 0.2676 | 0.1720 | 0.0370 | 0.0 | 0.0212 | 0.0646 | 0.0894 | 0.0128 | 5.0 |
| GDSS+TF | 0.0036 | 0.1206 | 0.0525 | 0.0137 | 5.0 | 0.0411 | 0.0565 | 0.0706 | 0.0074 | 27.5 |
| ConGress | 0.0048 | 0.2728 | 1.2950 | 0.0418 | 0.0 | 0.0273 | 0.1029 | 0.1148 | - | 0.0 |
| DiGress | **0.0003** | 0.0372 | **0.0009** | 0.0106 | 75. | 0.0013 | 0.0498 | 0.0434 | 0.0400 | 74.0 |
| GruM | 0.0005 | **0.0353** | **0.0009** | 0.0062 | 90.0 | **0.0007** | **0.0492** | 0.0448 | 0.0050 | **85.0** |
| **GBD** | **0.0003** | **0.0353** | 0.0135 | **0.0059** | **92.5** | 0.0013 | 0.0493 | 0.0446 | **0.0047** | 75.0 |

**Generating graphs with simple topology** Using the GBD theoretical model and the implementation detailed in Appendix E.1, we show that GBD excels at generating graphs with moderate size and simple topological patterns, as evidenced by the MMD metrics in Table 1. GBD outperforms the baselines in five out of nine experiments and remains statistically on par with the SOTA model in the other two. Even with larger graphs (*e.g.*, the `Grid` benchmark), GBD achieved a degree MMD of 0.045, significantly surpassing all baselines, as well as maintaining competitive performance on other metrics. Moreover, previous works (Cho et al., 2023; Jo et al., 2022; Liu et al., 2019) suggested that the MMDs as a metric suffers from large standard deviations due to the insufficient size of the sampled graphs. Thus, to gain a comprehensive study, we repeated the evaluation using an enlarged generated sample of 1024 graphs. As shown in Appendix F.2, the results demonstrate that GBD consistently generates graphs that resembles true data, with improved statistical significance in experimental conclusions.

**Generating graphs with complex topology** GBD also maintains a leading position in generating large graphs with complex topologies, as shown by the results in Table 2. In the experiments on the `Planar` and `SBM` datasets, GBD achieved superior or comparable MMD scores on most graph statistics, along with high V.U.N. scores, consistently ranking first or second among all baselines. These results provide stronger evidence of GBD's capability and suitability as a candidate model for generating graph data, particularly for applications where real-world instances are typically more complex in nature.

**Rapid convergence in the reverse process.** We conducted a convergence experiment measuring the V.U.N. metric for graphs $\mathbf{G}_t$ generated at every 100 steps on the reverse chain, with the results shown in Figure 4. The figure highlights that GBD achieves high V.U.N. score at early timestamps on the reverse diffusion process, reflecting rapid convergence. Notably, both GBD and the second-place model, GruM (Jo et al., 2023), share a key property of predicting $\mathbb{E}[\mathbf{G}_0 \mid \mathbf{G}_t]$. While GruM achieves

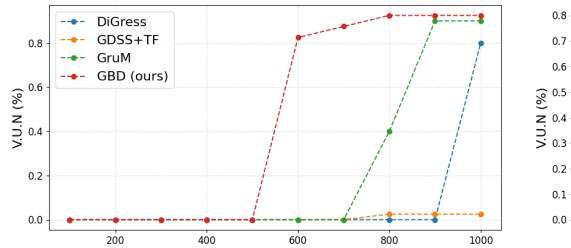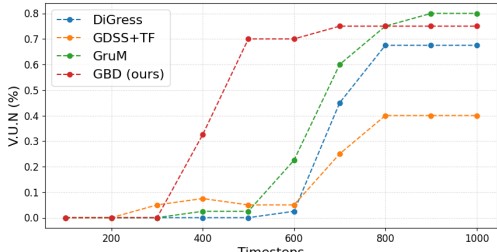

Figure 4: V.U.N. results for intermediate graph samples in the reverse chain on `Planar` (left) and `SBM` (right) datasets, with GBD demonstrating clear advantage in convergence rate.

Table 3: 2D molecule generation results

| Method | QM9 $(|N| \leq 9)$ | | | | ZINC250K $(|N| \leq 38)$ | | | |
|---|---|---|---|---|---|---|---|---|
| | Valid (%) ↑ | FCD ↓ | NSPDK ↓ | Scaf. ↑ | Valid(%) ↑ | FCD ↓ | NSPDK ↓ | Scaf. ↑ |
| MoFlow | 91.36 | 4.467 | 0.0169 | 0.1447 | 63.11 | 20.931 | 0.0455 | 0.0133 |
| GraphAF | 74.43 | 5.625 | 0.0207 | 0.3046 | 68.47 | 16.023 | 0.0442 | 0.0672 |
| GraphDF | 93.88 | 10.928 | 0.0636 | 0.0978 | 90.61 | 33.546 | 0.1770 | 0.0000 |
| EDP-GNN | 47.52 | 2.680 | 0.0046 | 0.3270 | 82.97 | 16.737 | 0.0485 | 0.0000 |
| GDSS | 95.72 | 2.900 | 0.0033 | 0.6983 | 97.01 | 14.656 | 0.0195 | 0.0467 |
| GDSS+TF | 99.68 | 0.737 | 0.0024 | 0.9129 | 96.04 | 5.556 | 0.0326 | 0.3205 |
| DiGress | 98.19 | 0.095 | 0.0003 | 0.9353 | 94.99 | 3.482 | 0.0021 | 0.4163 |
| SwinGNN | 99.71 | 0.125 | 0.0003 | - | 81.72 | 5.920 | 0.006 | - |
| GraphARM | 90.25 | 1.22 | 0.0020 | - | 88.23 | 16.26 | 0.055 | - |
| EDGE | 99.10 | 0.458 | - | 0.763 | - | - | - | - |
| GruM | 99.69 | 0.108 | **0.0002** | 0.9449 | **98.65** | 2.257 | **0.0015** | **0.5299** |
| **GBD** | **99.88** | **0.093** | **0.0002** | **0.9510** | 97.87 | **2.248** | 0.0018 | 0.5042 |

it through designing the model upon an OU bridge mixture (Jo et al., 2023), our model attain the same property due to the inherent nature of beta diffusion.

## 4.2 MOLECULE GENERATION

**Datasets and Metrics** We consider two widely-used molecule datasets as benchmarks in Jo et al. (2022): `QM9` (Ramakrishnan et al., 2014), which consists of 133,885 molecules with $N \leq 9$ nodes from 4 different node types, and `ZINC250k` (Irwin et al., 2012), which consists of 249,455 molecules with $N \leq 38$ nodes from 9 different node types. Molecules in both datasets have 3 edge types, namely single bond, double bond, and triple bond.

Following the evaluation setting of Jo et al. (2022), we generated 10,000 molecules for each dataset and evaluated them with four metrics: the ratio of valid molecules without correction (**Val.**), Fréchet ChemNet Distance (**FCD**), Neighborhood subgraph pairwise distance kernel **NSPDK**, and Scaffold similarity (**Scaf.**). We provide the results of uniqueness and novelty in Appendix E.2.

**Baselines** We compare GBD against the following autoregressive and one-shot graph generation methods: **MoFlow** (Zang & Wang, 2020), **GraphAF** (Shi et al., 2020), **GraphDF** (Luo et al., 2021), and several state-of-the-art diffusion-based graph generative models discussed previously: **EDP-GNN** (Niu et al., 2020a), **GDSS** (Jo et al., 2022), **ConGress** (Vignac et al., 2023), **DiGress** (Vignac et al., 2023), **SwinGNN** (Niu et al., 2020b), **GraphARM** (Kong et al., 2023), **EDGE** (Chen et al., 2023), and **GruM** (Jo et al., 2023). We describe the details of the implementation in Appendix E.2.

**Results** As shown in Table 3, we observe that our GBD outperforms most previous diffusion-based models and is competitive with the current state-of-the-art Gaussian-based diffusion model, GruM. In particular, compared to the basic continuous diffusion model, GBD significantly outperforms it (GDSS+TF) under the same GraphTransformer architecture. Additionally, we observe that our proposed beta-based graph diffusion model is superior to the discrete diffusion model on both 2D molecule datasets, demonstrating that our method is also capable of modeling complex structures of attributed graphs. We attribute this to the excellent modeling ability of the beta-based graph diffusion model for sparse data distributions.

### 4.3 ADDITIONAL EXPERIMENTAL RESULTS

**Ablation study on precondition and computation domain.** We vary the options regarding the computation domain and the application of preconditioning, and summarize the results in Table 4. The combination of adopting logit domain computation without using preconditioning can sometimes increase the challenge in model convergence, and therefore, it is not recommended. The listed results on `Ego-small` and `Community-small` demonstrate that both techniques are in general beneficial for achieving better model performance, and the effect of preconditioning is more evident when the computation is perfomed in the logit domain.

Table 4: The effect of logit domain computation and preconditioning

|                |                 | Ego-small | | | Community-small | | |
|----------------|-----------------|-------|-------|--------|-------|-------|--------|
| Logit Domain   | Preconditioning | Deg.  | Clus. | Orbit. | Deg.  | Clus. | Orbit. |
| -              | -               | 0.015 | 0.018 | 0.004  | 0.010 | 0.076 | 0.004  |
| -              | ✓               | 0.013 | 0.017 | **0.002** | 0.004 | **0.044** | 0.007 |
| ✓              | ✓               | **0.011** | **0.014** | **0.002** | **0.002** | 0.060 | **0.002** |

**Ablation study on concentration modulation.** To verify that effect of concentration modulation, we compared the results of GBD under different concentration modulation strategies on both the general graph and the molecule graph. As shown in Table 5, assigning the appropriate $\eta$ through optional concentration modulation strategies helps GBD model various types of graphs, and this technique has the potential for further modification according to varying scenarios.

Table 5: The effect of concentration modulation on generic graph and molecule graph.

| | Planar | | | | | SBM | | | | |
|----------------------|--------|--------|--------|--------|--------|--------|--------|--------|--------|--------|
| Modulation Strategy  | Deg.   | Clus.  | Orbit. | Spec.  | V.U.N. | Deg.   | Clus.  | Orbit. | Spec.  | V.U.N. |
| w/o modulation       | 0.0005 | 0.0357 | 0.0294 | 0.0069 | 87.5   | 0.0015 | 0.0493 | 0.0452 | 0.0051 | 72.5   |
| Betweenness Centrality | **0.0003** | 0.0354 | 0.0154 | **0.0057** | 90.0 | 0.0015 | **0.0492** | 0.0450 | 0.0053 | 70.0 |
| Degree Centrality    | **0.0003** | 0.0353 | 0.0135 | 0.0059 | **92.5** | **0.0013** | 0.0493 | **0.0446** | **0.0047** | **75.0** |

| | QM9 | | | | ZINC250K | | | |
|----------------------|----------------|---------|----------|----------|----------------|---------|----------|----------|
| Modulation Strategy  | Valid(%) ↑     | FCD ↓   | NSPDK ↓  | Spec. ↓  | Valid(%) ↑     | FCD ↓   | NSPDK ↓  | Spec. ↓  |
| w/o modulation       | 99.73          | 0.126   | 0.0003   | 0.9475   | 97.60          | 2.412   | 0.0020   | 0.5033   |
| Carbon Bonds         | **99.88**      | **0.093** | **0.0002** | **0.9510** | **97.87**    | **2.248** | **0.0018** | **0.5042** |

**Comparison on various node feature initialization.** To demonstrate that GBD has the potential to model graphs with various types of node features, we explored the impact of different initialization of node features on modeling the joint distribution $\mathbf{G} = (\mathbf{A}, \mathbf{X})$. Specifically, the node representation can be featured by Degree, Centrailities, and Eigenvectors and we vary the node feature initialization and summarize the results of model performance, as detailed in Appendix F.2.

### 4.4 VISUALIZATION

We present samples from GBD trained on `planar`, `SBM`, `QM9` and `ZINC250k` in Figure 3. Additionally, we provide visualization of the generative process and more generated graphs of GBD, along with a comprehensive description in Appendix G.

## 5 CONCLUSION

We introduce graph beta diffusion (GBD), a novel graph generation framework developed upon beta diffusion. We demonstrate that the utilization of beta distribution to define the diffusion process is beneficial for modeling the distribution of graph data, and outline four crucial designing elements—data transformation, concentration modulation, logit-domain computation, and neural-network precondition—that consistently enhance model performance. Through extensive experiments, GBD demonstrated superior performance across various graph benchmarks, showcasing its ability to model diverse patterns within graph data. Moreover, our proposed model shows promise in modeling various types of data with discrete structures and offers valuable insights into further exploring the properties of beta diffusion.

## REPRODUCIBILITY STATEMENT

Our PyTorch code is available at https://github.com/xinyangATK/GraphBetaDiffusion. The implementation follows the training and sampling algorithms described in Appendix D.1 and other technical details outlined in Appendix D.

## ACKNOWLEDGMENT

M. Zhou acknowledges the support of a gift fund from Apple.

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

## A  BACKGROUND OF BETA DIFFUSION MODEL

In this section, we employ $x$ to denote the original data that is already normalized to the range $[0, 1]$. The variables $z_1$ through $z_T$ denote the latent states sampled via a forward beta diffusion process, with larger subscripts indicating states closer to complete noise. The beta diffusion model (Zhou et al., 2023) is designed so that each latent state follows a beta distribution as its conditional marginal distribution:

$$q(z_t \mid z_0) = \text{Beta}(z_t \mid \eta\alpha_t z_0, \eta(1 - \alpha_t)z_0). \tag{19}$$

In this setup, $\eta$ serves as the concentration parameter, where larger values of $\eta$ result in a stronger concentration effect, shaping the beta distribution more sharply around its mean. The parameter $\alpha_t$ is a scheduling factor that gradually decreases from 1 to 0 as the forward diffusion process advances, controlling the degree of noise introduced at each step.

To achieve this, the transition between two consecutive states in the forward diffusion process is formulated as a multiplication by an attenuating factor, *i.e.*,

$$z_t = z_{t-1} \cdot q_t, \ q_t \sim \text{Beta}(\eta\alpha_t z_0, \eta(\alpha_{t-1} - \alpha_t)), \ t \in [1, T]. \tag{20}$$

To demonstrate that the forward process defined via Equation 20 results in the conditional marginal distribution given in Equation 19, one can utilize the property that

$$(z_t, z_{t-1} - z_t, 1 - z_{t-1}) \sim \text{Dir}(\eta\alpha_t z_0, \eta(\alpha_{t-1} - \alpha_t)z_0, \eta(1 - \alpha_{t-1}z_0)), \ t \in [1, T] \tag{21}$$

and then apply mathematical induction to establish the desired conclusion.

It is worth noting that a generic triplet $(\pi_1, \pi_2, \pi_3)$ from the Dirichlet distribution in Equation 21 can be generated using the stick-breaking construction. Specifically, $\pi_1$ is first sampled from $\text{Beta}(\eta\alpha_t z_0, \eta(1 - \alpha_t z_0))$. Next, $\pi_2$ is created by multiplying $(1 - \pi_1)$ by a factor sampled from $\text{Beta}(\eta(\alpha_{t-1} - \alpha_t)z_0, \eta(1 - \alpha_{t-1}z_0))$. Finally, $\pi_3$ can be obtained by subtracting $(\pi_1 + \pi_2)$ from one. This aligns with the reverse transition process: once $z_t$ is obtained, $z_{t-1}$ can be reconstructed by adding an increment, generated in the same manner as $\pi_2$, to $z_t$, *i.e.*,

$$z_{t-1} = z_t + (1 - z_t) \cdot p_t, \ p_t \sim \text{Beta}(\eta(\alpha_{t-1} - \alpha_t)z_0, \eta(1 - \alpha_{t-1}z_0)), \ t \in [1, T]. \tag{22}$$

With the reverse transition process formalized in Equation 22, one can derive the reverse transition distribution as

$$q(z_{t-1} \mid z_t, z_0) = \frac{1}{1 - z_t}\text{Beta}\left(\frac{z_{t-1} - z_t}{1 - z_{t-1}} \mid \eta(\alpha_{t-1} - \alpha_t)z_0, \eta(1 - \alpha_{t-1}z_0)\right), t \in [1, T]. \tag{23}$$

In practice, to implement the reverse diffusion process, Zhou et al. (2023) defines the ancestral sampling distribution as following:

$$p_\theta(z_{t-1} \mid z_t) := q(z_{t-1} \mid z_t, f_\theta(z_t, t)), \tag{24}$$

with $f_\theta$ denoting a neural network with parameters $\theta$, that predicts $z_0$ with latent state $z_t$ and time stamp $t$.

## B  ANALYTICAL EXPRESSIONS OF OPTIMIZATION OBJECTIVE

Recall that the optimization function is expressed as

$$\mathcal{L} = \sum_{t=2}^{T}(1 - \omega)\mathcal{L}_{\text{sampling}}(t, \mathbf{G}_0) + \omega\,\mathcal{L}_{\text{correction}}(t, \mathbf{G}_t), \ \omega \in [0, 1], \tag{25}$$

with the components defined as follows:

$$\mathcal{L}_{\text{sampling}}(t, \mathbf{G}_0) \triangleq \mathbb{E}_{q(\mathbf{G}_t, \mathbf{G}_0)}\,\text{KL}\left(p_\theta(\mathbf{G}_{t-1} \mid \mathbf{G}_t) \parallel q(\mathbf{G}_{t-1} \mid \mathbf{G}_t, \mathbf{G}_0)\right), \tag{26}$$

$$\mathcal{L}_{\text{correction}}(t, \mathbf{G}_0) \triangleq \mathbb{E}_{q(\mathbf{G}_t, \mathbf{G}_0)}\,\text{KL}\left(q(\mathbf{G}_\tau \mid \hat{G}_\theta(\mathbf{G}_t, t)) \parallel q(\mathbf{G}_\tau \mid \mathbf{G}_0)\right). \tag{27}$$

To derive the analytical form of $\mathcal{L}_{\text{sampling}}$ and $\mathcal{L}_{\text{correction}}$, we employ the following property of beta distribution (Joo et al., 2020; Zhou et al., 2023):

**Property 2** *The KL divergence between two Beta distributions* $\mathrm{Beta}(\alpha_1, \beta_1)$ *and* $\mathrm{Beta}(\alpha_2, \beta_2)$ *is given by:*

$$KL\left(\mathrm{Beta}(\alpha_1, \beta_1) \,\|\, \mathrm{Beta}(\alpha_2, \beta_2)\right)$$

$$= \ln \frac{B(\alpha_2, \beta_2)}{B(\alpha_1, \beta_1)} + (\alpha_1 - \alpha_2)\left[\psi(\alpha_1) - \psi(\alpha_1 + \beta_1)\right] + (\beta_1 - \beta_2)\left[\psi(\beta_1) - \psi(\alpha_1 + \beta_1)\right], \quad (28)$$

*where* $B(\alpha, \beta) \overset{\Delta}{=} \frac{\Gamma(\alpha)\Gamma(\beta)}{\Gamma(\alpha+\beta)}$ *is the Beta function,* $\psi(\cdot)$ *denotes the digamma function.*

With probability distributions $q(\mathbf{G}_{t-1} \mid \mathbf{G}_t, \mathbf{G}_0)$ and $p_\theta(\mathbf{G}_{t-1} \mid \mathbf{G}_t)$ defined in Equations 6 and 7, the parameters $(\alpha_p, \beta_p, \alpha_q, \beta_q)$ are instantiated as follows:

$$\alpha_1 = \eta(\alpha_{t-1} - \alpha_t)\hat{\mathbf{G}}_0, \;\; \beta_1 = \eta(1 - \alpha_{t-1}\hat{\mathbf{G}}_0),$$
$$\alpha_2 = \eta(\alpha_{t-1} - \alpha_t)\mathbf{G}_0, \;\; \beta_2 = \eta(1 - \alpha_{t-1}\mathbf{G}_0).$$

Substituting these into Equation 28, we can express the KL term within $\mathcal{L}_{\mathrm{sampling}}$ as

$$\begin{aligned}
&KL\left(p_\theta(\mathbf{G}_{t-1} \mid \mathbf{G}_t) \,\|\, q(\mathbf{G}_{t-1} \mid \mathbf{G}_t, \mathbf{G}_0)\right) \\
&= \ln\Gamma(\eta(\alpha_{t-1} - \alpha_t)\mathbf{G}_0) + \ln\Gamma(\eta - \eta\alpha_{t-1}\mathbf{G}_0) - \ln\Gamma(\eta - \eta\alpha_t\mathbf{G}_0) \\
&\quad - \ln\Gamma(\eta(\alpha_{t-1} - \alpha_t)\hat{\mathbf{G}}_0) - \ln\Gamma(\eta - \eta\alpha_{t-1}\hat{\mathbf{G}}_0) + \ln\Gamma(\eta - \eta\alpha_t\hat{\mathbf{G}}_0) \\
&\quad + \eta(\alpha_{t-1} - \alpha_t)(\hat{\mathbf{G}}_0 - \mathbf{G}_0) \cdot \psi(\eta(\alpha_{t-1} - \alpha_t)\hat{\mathbf{G}}_0) \\
&\quad + \eta\alpha_{t-1}(\mathbf{G}_0 - \hat{\mathbf{G}}_0) \cdot \psi(\eta - \eta\alpha_{t-1}\hat{\mathbf{G}}_0) \\
&\quad + \eta\alpha_t(\hat{\mathbf{G}}_0 - \mathbf{G}_0) \cdot \psi(\eta - \eta\alpha_t\hat{\mathbf{G}}_0),
\end{aligned} \quad (29)$$

where $\hat{\mathbf{G}}_0 := \hat{G}_\theta(\mathbf{G}_t, t)$.

Similarly, the KL term within $\mathcal{L}_{\mathrm{correction}}$ can be derived as

$$\begin{aligned}
&KL\left(q(\mathbf{G}_\tau \mid \hat{G}_\theta(\mathbf{G}_t, t)) \,\|\, q(\mathbf{G}_\tau \mid \mathbf{G}_0)\right) \\
&= \ln\Gamma(\eta\alpha_t\mathbf{G}_0) + \ln\Gamma(\eta - \eta\alpha_t\mathbf{G}_0) - \ln\Gamma(\eta\alpha_t\hat{\mathbf{G}}_0) - \ln\Gamma(\eta - \eta\alpha_t\hat{\mathbf{G}}_0) \\
&\quad + \eta\alpha_t(\hat{\mathbf{G}}_0 - \mathbf{G}_0) \cdot \left(\psi(\eta\alpha_t\hat{\mathbf{G}}_0) - \psi(\eta - \eta\alpha_t\hat{\mathbf{G}}_0)\right).
\end{aligned} \quad (30)$$

## C  MATHEMATICAL EXPRESSIONS FOR ELEMENTS IN NEURAL-NETWORK PRECONDITIONING

In this section, we present the full derivation of the expressions for the mean and variance of $a_t$ and $x_t$ as shown in Equations 17 and 18, and extend these conclusions to $\mathrm{logit}(a_t)$ and $\mathrm{logit}(x_t)$. To establish these results, we begin by introducing the following property of the beta distribution:

**Property 3** *Given that* $g_t \mid g_0 \sim \mathrm{Beta}(\eta\alpha_t g_0, \eta(1 - \alpha_t g_0))$, *one can derive that* $\mathbb{E}[g_t \mid g_0] = \alpha_t g_0$ *and* $\mathrm{Var}[g_t \mid g_0] = \frac{\alpha_t g_0(1 - \alpha_t g_0)}{\eta + 1}$. *Let* $\mu := \mathbb{E}[g_0]$ *and* $\sigma^2 := \mathrm{Var}[g_0]$. *By applying the law of total expectation and the law of total variance, we obtain the following results:*

$$\mathbb{E}[g_t] = \alpha_t\mu, \;\; \mathrm{Var}[g_t] = \frac{\alpha_t\mu - \alpha_t^2(\mu^2 + \sigma^2)}{\eta + 1} + \alpha_t^2\sigma^2. \quad (31)$$

*Their counterparts in the logit domain are expressed as*

$$\mathbb{E}[\mathrm{logit}(g_t)] = \mathbb{E}[\psi(\eta\alpha_t g_0)] - \mathbb{E}[\psi(\eta - \eta\alpha_t g_0)], \quad (32)$$

$$\begin{aligned}
\mathrm{Var}[\mathrm{logit}(g_t)] &= \mathbb{E}[\psi^{(1)}(\eta\alpha_t g_0)] + \mathbb{E}[\psi^{(1)}(\eta(1 - \alpha_t g_0))] \\
&\quad + \mathrm{Var}[\psi(\eta\alpha_t g_0)] + \mathrm{Var}[\psi(\eta(1 - \alpha_t g_0))],
\end{aligned} \quad (33)$$

*with* $\psi(\cdot)$ *and* $\psi^{(1)}(\cdot)$ *denoting digamma and trigamma functions.*

In the example presented in Section 2.3, we assume that $a_0$ follows a categorical distribution with outcomes $a_{\min}$ and $a_{\max}$, where the probability of $P(a_0 = a_{\max}) = p$. This leads to the expected

value $\mu = p \cdot a_{\max} + (1 - p) \cdot a_{\min}$ and variance $\sigma^2 = p(1 - p)(a_{\max} - a_{\min})^2$. Taking these quantities into Equation 31, we obtain

$$\mathbb{E}[a_t] = \alpha_t \left( p \cdot a_{\max} + (1 - p) \cdot a_{\min} \right),$$

$$\mathrm{Var}[a_t] = \frac{1}{\eta_A + 1} \left( \mathbb{E}[a_t] - \mathbb{E}[a_t]^2 \right) + \frac{\eta_A}{\eta_A + 1} \left( \alpha_t^2 (p(1 - p))(a_{\max} - a_{\min})^2 \right).$$

For node features, the assumption states that $x_0$ is uniformly distributed over the interval $[x_{\min}, x_{\max}]$. This gives a mean of $\mu = (x_{\min} + x_{\max})/2$ and a variance of $\sigma^2 = (x_{\max} - x_{\min})^2/12$, and the corresponding mean and variance of $x_t$ can then be expressed as

$$\mathbb{E}[x_t] = \frac{1}{2}\alpha_t(x_{\min} + x_{\max}),$$

$$\mathrm{Var}[x_t] = \frac{1}{\eta_X + 1} \left( \mathbb{E}[x_t] - \mathbb{E}[x_t]^2 \right) + \frac{\eta_X}{12(\eta_X + 1)} \left( \alpha_t^2 (x_{\max} - x_{\min})^2 \right).$$

Under the same probabilistic assumptions for $a_0$ and $x_0$, we can derive the expressions for the terms in Equations 32 and 33, presented in the following remarks:

**Remark 1** *Given that $a_0$ has two potential outcomes $\{a_{\min}, a_{\max}\}$ with $P(a_0 = a_{\max}) = p$, the computation of $\mathbb{E}[\mathrm{logit}(g_t)]$ and $\mathrm{Var}[\mathrm{logit}(g_t)]$ involves several key components, including:*

$$\mathbb{E}[\psi(\eta\alpha_t a_0)] = p \cdot \psi(\eta\alpha_t a_{\max}) + (1 - p) \cdot \psi(\eta\alpha_t a_{\min}),$$
$$\mathbb{E}[\psi(\eta - \eta\alpha_t a_0)] = p \cdot \psi(\eta - \eta\alpha_t a_{\max}) + (1 - p) \cdot \psi(\eta - \eta\alpha_t a_{\min}),$$
$$\mathrm{Var}[\psi(\eta\alpha_t a_0)] = p(1 - p) \left( \psi(\eta\alpha_t a_{\max}) - \psi(\eta\alpha_t a_{\min}) \right)^2,$$
$$\mathrm{Var}[\psi(\eta - \eta\alpha_t a_0)] = p(1 - p) \left( \psi(\eta - \eta\alpha_t a_{\max}) - \psi(\eta - \eta\alpha_t a_{\min}) \right)^2,$$
$$\mathbb{E}[\psi^{(1)}(\eta\alpha_t a_0)] = p \cdot \psi^{(1)}(\eta\alpha_t a_{\max}) + (1 - p) \cdot \psi^{(1)}(\eta\alpha_t a_{\min}),$$
$$\mathbb{E}[\psi^{(1)}(\eta - \eta\alpha_t a_0)] = p \cdot \psi^{(1)}(\eta - \eta\alpha_t a_{\max}) + (1 - p) \cdot \psi^{(1)}(\eta - \eta\alpha_t a_{\min}). \tag{34}$$

**Remark 2** *Let $x_0$ be uniformly distributed as $\mathrm{Unif}[x_{\min}, x_{\max}]$. We denote $K$ as the number of sub-intervals used for numerical integration via the Trapezoidal rule. Similar to Remark 1, we present the expressions for the components in the logit domain as follows:*

$$\mathbb{E}[\psi(\eta\alpha_t x_0)] = \frac{1}{\eta\alpha_t(x_{\max} - x_{\min})} \left( \ln\Gamma(\eta\alpha_t x_{\max}) - \ln\Gamma(\eta\alpha_t x_{\min}) \right),$$

$$\mathbb{E}[\psi(\eta - \eta\alpha_t x_0)] = \frac{1}{\eta\alpha_t(x_{\max} - x_{\min})} \left( \ln\Gamma(\eta - \eta\alpha_t x_{\min}) - \ln\Gamma(\eta - \eta\alpha_t x_{\max}) \right),$$

$$\mathrm{Var}[\psi(\eta\alpha_t x_0)] \approx \max\left( \frac{1}{K} \sum_{i=0}^{K} \frac{\psi^2\left(\eta\alpha_t\left(x_{\min} + \frac{i}{K}(x_{\max} - x_{\min})\right)\right)}{2^{\delta(i=0)+\delta(i=K)}} - \mathbb{E}[\psi(\eta\alpha_t x_0)]^2, 0 \right),$$

$$\mathrm{Var}[\psi(\eta - \eta\alpha_t x_0)] \approx \max\left( \frac{1}{K} \sum_{i=0}^{K} \frac{\psi^2\left(\eta - \eta\alpha_t\left(x_{\min} + \frac{i}{K}(x_{\max} - x_{\min})\right)\right)}{2^{\delta(i=0)+\delta(i=K)}} - \mathbb{E}[\psi(\eta - \eta\alpha_t x_0)]^2, 0 \right),$$

$$\mathbb{E}[\psi^{(1)}(\eta\alpha_t x_0)] = \frac{1}{\eta\alpha_t(x_{\max} - x_{\min})} \left( \psi(\eta\alpha_t x_{\max}) - \psi(\eta\alpha_t x_{\min}) \right),$$

$$\mathbb{E}[\psi^{(1)}(\eta - \eta\alpha_t x_0)] = \frac{1}{\eta\alpha_t(x_{\max} - x_{\min})} \left( \psi(\eta - \eta\alpha_t x_{\min}) - \psi(\eta - \eta\alpha_t x_{\max}) \right). \tag{35}$$

# D DETAILS OF GBD

## D.1 TRAINING AND SAMPLING

We provide the pseudo-code of the training and sampling of our generative framework within original domain and logit domain, respectively. Specifically, Algorithm 3 and Algorithm 1 show the procedure of training and sampling in original domain, respectively. In practice, we migrate our proposed GBD to logit domain shown in Algorithm 4 and Algorithm 2 in most cases.

**Algorithm 1** Sampling, in original domain.

---

**Require:** Number of time steps $T = 1000$, default concentration parameter $\eta = 30$, predictor $\hat{G}_\theta$.
1: (Optional) Assign value to $\eta$ via Equation 38
2: Sample $\mathbf{G}_T = (\mathbf{A}_T, \mathbf{X}_T) \sim p(\mathbf{A}_T, \mathbf{X}_T)$
3: **for** $t = T$ to 1 **do**
4:      $\mathbf{G}_{in} = \frac{\mathbf{G}_t - \mathbb{E}[\mathbf{G}_t]}{\sqrt{\mathrm{Var}[\mathbf{G}_t]}}$
5:      $(\hat{\mathbf{A}}_0', \hat{\mathbf{X}}_0') \leftarrow \hat{G}_\theta(\mathbf{G}_{in}, t)$
6:      $\hat{\mathbf{A}}_0 \leftarrow w_A \cdot \hat{\mathbf{A}}_0' + b_A$
7:      $\hat{\mathbf{X}}_0 \leftarrow w_X \cdot \hat{\mathbf{X}}_0' + b_X$
8:      $\hat{\mathbf{G}}_0 \leftarrow (\hat{\mathbf{A}}_0, \hat{\mathbf{X}}_0)$
9:      $\mathbf{P}_t \sim \mathrm{Beta}(\eta(\alpha_{t-1} - \alpha_t)\hat{\mathbf{G}}_0, \eta(1 - \alpha_{t-1}\hat{\mathbf{G}}_0))$
10:      $\mathbf{G}_{t-1} \leftarrow \mathbf{G}_t + \mathbf{P}_t \odot (1 - \mathbf{G}_t)$
11: **end for**
12: **return** $(\mathbf{A}_0 - b_A)/w_A$ and $(\mathbf{X}_0 - b_X)/w_X$

---

**Algorithm 2** Sampling, in logit domain

---

1: Sample $\mathrm{logit}(\mathbf{G}_T) \sim p(\mathrm{logit}(\mathbf{A}_T), \mathrm{logit}(\mathbf{X}_T))$
2: **for** $t = 1$ to 1 **do**
3:      $\mathbf{G}_{in} = \frac{\mathrm{logit}(\mathbf{G}_t) - \mathbb{E}[\mathrm{logit}(\mathbf{G}_t)]}{\sqrt{\mathrm{Var}[\mathrm{logit}(\mathbf{G}_t)]}}$
4:      $(\hat{\mathbf{A}}_0', \hat{\mathbf{X}}_0') \leftarrow \hat{G}_\theta(\mathbf{G}_{in}, t)$
5:      $\hat{\mathbf{A}}_0 \leftarrow w_A \cdot \hat{\mathbf{A}}_0' + b_A$
6:      $\hat{\mathbf{X}}_0 \leftarrow w_X \cdot \hat{\mathbf{X}}_0' + b_X$
7:      $\hat{\mathbf{G}}_0 \leftarrow (\hat{\mathbf{A}}_0, \hat{\mathbf{X}}_0)$
8:      $\mathbf{U}_t \sim \mathrm{Gamma}(\eta(\alpha_{t-1} - \alpha_t)\hat{\mathbf{G}}_0, 1)$
9:      $\mathbf{V}_t \sim \mathrm{Gamma}(\eta(1 - \alpha_{t-1}\hat{\mathbf{G}}_0), 1)$
10:      $\mathrm{logit}(\mathbf{P}_t) \leftarrow \ln \mathbf{U}_t - \ln \mathbf{V}_t$
11:      Obtain $\mathrm{logit}(\mathbf{G}_{t-1})$ from Equation 15
12: **end for**
13: $\mathbf{G}_0 \leftarrow \mathrm{sigmoid}(\mathrm{logit}(\mathbf{G}_0))$
14: **return** $(\mathbf{A}_0 - b_A)/w_A$ and $(\mathbf{X}_0 - b_X)/w_X$

---

**Algorithm 3** Training, in original domain.

---

**Require:** Number of timesteps $T = 1000$, default concentration parameter $\eta = 30$, predictor $\hat{G}_\theta$, default node influence factor $\gamma = 0.5$, input graph batch $\mathbb{B} = \{\mathbf{G}^{(k)} = (\mathbf{A}^{(k)}, \mathbf{X}^{(k)})\}_{[K]}$, default learning rate $\lambda = 0.002$, optimization steps $M$.
1: (Optional) Assign value to $\eta$ via Equation 38
2: **for** step $= 1$ to $M$ **do**
3:      Initialize $\mathcal{L}_X$ and $\mathcal{L}_A$ with 0
4:      **for** $k = 1$ to $K$ **do**
5:          $t \sim \mathrm{Unif}(1, ..., T)$
6:          $\alpha_t, \alpha_{t-1} \leftarrow \mathrm{schedule}(t), \mathrm{schedule}(t-1)$
7:          $\mathbf{A}_0 \leftarrow w_A \cdot \mathbf{A}^{(k)} + b_A$
8:          $\mathbf{X}_0 \leftarrow w_X \cdot \mathbf{X}^{(k)} + b_X$
9:          $\mathbf{G}_0 \leftarrow (\mathbf{A}_0, \mathbf{X}_0)$
10:          $\mathbf{G}_t \sim \mathrm{Beta}(\eta\alpha_t\mathbf{G}_0, \eta(1 - \alpha_t\mathbf{G}_0))$
11:          $\mathbf{G}_{in} \leftarrow \frac{\mathbf{G}_t - \mathbb{E}[\mathbf{G}_t]}{\sqrt{\mathrm{Var}[\mathbf{G}_t]}}$
12:          $(\hat{\mathbf{A}}_0', \hat{\mathbf{X}}_0') \leftarrow \hat{G}_\theta(\mathbf{G}_{in}, t)$
13:          $\hat{\mathbf{A}}_0 = w_A \cdot \hat{\mathbf{A}}_0' + b_A$
14:          $\hat{\mathbf{X}}_0 = w_X \cdot \hat{\mathbf{X}}_0' + b_X$
15:          $\mathcal{L}_A \leftarrow \mathcal{L}_A + \mathcal{L}(\mathbf{A}_0, \hat{\mathbf{A}}_0, \eta, \alpha_t, \alpha_{t-1})$
16:          $\mathcal{L}_X \leftarrow \mathcal{L}_X + \mathcal{L}(\mathbf{X}_0, \hat{\mathbf{X}}_0, \eta, \alpha_t, \alpha_{t-1})$
17:      **end for**
18:      $\theta \leftarrow \theta - \frac{\lambda}{K} \nabla_\theta(\mathcal{L}_A + \gamma\mathcal{L}_X)$
19: **end for**

---

**Algorithm 4** Training, in logit domain.

---

**Require:** Number of timesteps $T$, concentration parameter $\eta$, predictor $\hat{G}_\theta$, node influence factor $\gamma$, input graph batch $\mathbb{B}$, learning rate $\lambda$, optimization steps $M$. Same default values with Algorithm 3.
1: **for** step $= 1$ to $M$ **do**
2:      Initialize $\mathcal{L}_X$ and $\mathcal{L}_A$ with 0
3:      **for** $k = 1$ to $K$ **do**
4:          $t \sim \mathrm{Unif}(1, ..., T)$
5:          $\alpha_t, \alpha_{t-1} \leftarrow \mathrm{schedule}(t), \mathrm{schedule}(t-1)$
6:          $\mathbf{A}_0 \leftarrow w_A \cdot \mathbf{A}^{(k)} + b_A$
7:          $\mathbf{X}_0 \leftarrow w_X \cdot \mathbf{X}^{(k)} + b_X$
8:          $\mathbf{G}_0 \leftarrow (\mathbf{A}_0, \mathbf{X}_0)$
9:          $\mathbf{U}_t \sim \mathrm{Gamma}(\eta\alpha_t\mathbf{G}_0, 1)$
10:          $\mathbf{V}_t \sim \mathrm{Gamma}(\eta(1 - \alpha_t\mathbf{G}_0), 1)$
11:          $\mathrm{logit}(\mathbf{G}_t) \leftarrow \ln \mathbf{U}_t - \ln \mathbf{V}_t$
12:          $\mathbf{G}_{in} \leftarrow \frac{\mathrm{logit}(\mathbf{G}_t) - \mathbb{E}[\mathrm{logit}(\mathbf{G}_t)]}{\sqrt{\mathrm{Var}[\mathrm{logit}(\mathbf{G}_t)]}}$
13:          $(\hat{\mathbf{A}}_0', \hat{\mathbf{X}}_0') \leftarrow \hat{G}_\theta(\mathbf{G}_{in}, t)$
14:          $\hat{\mathbf{A}}_0 = w_A \cdot \hat{\mathbf{A}}_0' + b_A$
15:          $\hat{\mathbf{X}}_0 = w_X \cdot \hat{\mathbf{X}}_0' + b_X$
16:          $\mathcal{L}_A \leftarrow \mathcal{L}_A + \mathcal{L}(\mathbf{A}_0, \hat{\mathbf{A}}_0, \eta, \alpha_t, \alpha_{t-1})$
17:          $\mathcal{L}_X \leftarrow \mathcal{L}_X + \mathcal{L}(\mathbf{X}_0, \hat{\mathbf{X}}_0, \eta, \alpha_t, \alpha_{t-1})$
18:      **end for**
19:      $\theta \leftarrow \theta - \frac{\lambda}{K} \nabla_\theta(\mathcal{L}_A + \gamma\mathcal{L}_X)$
20: **end for**

---

## D.2 DETAILS OF CONCENTRATION MODULATION

Here we elaborate the concentration modulation strategies mentioned in Section 2.3.

**Concentration Modulation for general graph generation** For general graph generation, we provide two strategies depending on node-level centralities, which are degree centrality and betweenness centrality, respectively. For the degree centrality of the node $u$ in an undirected graph, it can be formulated as

$$C_d(u) = C_{in}(u) = C_{out}(u) = \mathrm{Deg}(u) \tag{36}$$

where $C_{in}(u)$, $C_{out}(u)$ and $\mathrm{Deg}(u)$ are denoted as the in-degree, out-degree, and total degree of the node $u$. For the betweenness centrality of the node $u$ in an undirected graph, it can be formulated as

$$C_b(u) = \sum_{s \neq t \neq u} \frac{g_{st}(u)}{g_{st}} \qquad (37)$$

where $g_{st}$ is the total number of shortest paths from node $s$ to node $t$, $g_{st}(u)$ is the number of those paths that pass through $v$. For large graphs, exact calculation of betweenness centrality can be time-consuming, thus approximation algorithms using random sampling are often employed. In practice, we utilize the library of NetworkX Hagberg et al. (2008) to implement this.

With these two metrics to measure the centrality of the nodes, which are denoted as $C(u)$ in general, the modulated $\eta$ can be mathematically expressed as

$$\eta_{u,v} = g_A(\max(\mathrm{C}(u), \mathrm{C}(v))), \quad \eta_u = g_X(\mathrm{C}(u)). \qquad (38)$$

where $g_A(\cdot)$ and $g_X(\cdot)$ are two assignment functions that map the node centrality to one of the predefined $\eta$ values.

**Concentration Modulation for molecule graph generation**   For molecule graph generation, we provide a straightforward strategy that regards the carbon atom as the most important node, as well as the bonds connected to the carbon atoms as the important edge in a molecule graph. Specifically, for various types of carbon-atom bond, we first rank the importance of bonds in the following order: carbon-carbon bonds, carbon-nitrogen bonds, carbon-oxygen bonds, carbon-sulfur bonds, and so on. Then we can assign predefined $\eta$ on different nodes and edges depending on their "importance" in a molecule graph.

### D.3   MODEL ARCHITECHTURE

We leverage the graph transformer network introduced in Dwivedi & Bresson (2020) and Vignac et al. (2023) cross all graph generation tasks. Each graph transformer layer consists of a graph attention module, as well as fully-connected layers and layer normalization. It employs self-attention module to update node features, then uses FiLM layers (Perez et al., 2018) to incorporate edge features and global features. Since the data we transformed falls within the range of [0, 1], we apply the sigmoid function to the output of node features and adjacency matrices to model the one-hot encoded representation of node and edge.

### D.4   SCHEDULE OF DIFFUSION PROCESS IN GBD

Following Zhou et al. (2023), we employ a sigmoid diffusion schedule defined as $\alpha_t = 1/(1 + e^{-c_0 - (c_1 - c_0)^t})$ throughout all experiments, where $c_0 = 10$ and $c_1 = -13$.

## E   EXPERIMENTAL DETAILS

### E.1   GENERAL GRAPH GENERATION

**Datasets**   We evaluated our model using three synthetic and real datasets of varying size and connectivity, previously used as benchmarks in the literature (Cho et al., 2023; Jo et al., 2022): Ego-small (Sen et al., 2008) consists of 200 small real sub-graphs from the Citeseer network dataset with $4 \leq N \leq 18$. Community-small consists of 100 randomly generated synthetic graphs with $12 \leq N \leq 20$, where the graphs are constructed by two equal-sized communities, each of which is generated by the Erdös–Rényi model (Erdős et al., 1960), with $p = 0.7$ and $0.05N$ inter-community edges are added with uniform probability as in previous works (Jo et al., 2022; Niu et al., 2020b). Grid consists of randomly generated 100 standard 2D grid graphs with $100 \leq N \leq 400$ and the maximum number of edges per node is 4 since all nodes are arranged in a regular lattice. Planar comprises 200 synthetic planar graphs, each containing $N = 64$. nodes. SBM includes 200 synthetic stochastic block model graphs, where the number of communities is randomly sampled from 2 to 5, and the number of nodes in each community is randomly sampled from 20 to 40. The probability of edges between communities is 0.3 and that of edges within communities is 0.05.

**Evaluation metrics** For a fair comparison, we follow the experimental and evaluation settings of Jo et al. (2022; 2023), using the same train/test split, where 80% of the data is used as the training set and the remaining 20% as the test set. We adopt maximum mean discrepancy (MMD) as our evaluation metric to compare three graph property distributions between test graphs and the same number of generated graphs: degree (**Deg.**), clustering coefficient (**Clus.**) and count of orbits with 4 nodes (**Orbit**). Note that we use the Gaussian Earth Mover's Distance (EMD) kernel to compute the MMDs following the method used in previous work (Jo et al., 2022; Cho et al., 2023). Additionally, for graphs with more complex structures like `Planar` and `SBM`, we report the eigenvalues of the graph Laplacian (**Spec.**) and the percentage of valid, unique, and novel graphs (**V.U.N.**) to measure whether the model has learned the intrinsic feature distribution and global properties of the graph. A lower value is better for all of these metrics except V.U.N. Specifically, a graph is defined as a valid planar graph if it is connected and planar. Following the statistical test introduced in Martinkus et al. (2022), we determine that a graph is a valid SBM graph if and only if it has a community count between 2 and 5, and a node count inside each community between 20 and 40.

**Implementation details** We follow the evaluation setting of Jo et al. (2022); Cho et al. (2023) to generate graphs of the same size as the test data in each run and we report the mean and standard deviation obtained from 3 independent runs for each dataset. We report the baseline results taken from Cho et al. (2023), except for the results of ConGress in Tables 1 and 8, which we obtained by running its corresponding open-source code. For a fair comparison, we adopt the Graph Transformer (Dwivedi & Bresson, 2020; Vignac et al., 2023) as the neural network used in GDSS+Transformer (Jo et al., 2022), DiGress (Vignac et al., 2023), and DruM (Jo et al., 2023). We set the diffusion steps to 1000 for all the diffusion models. For important hyperparameters mentioned in Sec 2.3, we usually set $S_{cale} = 0.9$, $S_{hift} = 0.09$. and $\eta = [10000, 100, 30, 10]$ for the normalized degrees corresponding to the intervals falling in the interval split by $[1.0, 0.8, 0.4, 0.1]$, respectively. In practice, we set threshold as 0.5 to quantize generated continue adjacency matrix.

### E.2 2D MOLECULE GENERATION

**Datasets** We utilize two widely-used molecular datasets as benchmarks, as described in Jo et al. (2023): **QM9** (Ramakrishnan et al., 2014), consisting of 133,885 molecules with $N \leq 9$ nodes from 4 different node types and **ZINC250k** (Irwin et al., 2012), consisting of 249,455 molecules with $N \leq 38$ nodes from 9 node types. Molecules in both datasets have 3 edge types, namely single bond, double bond, and triple bond. Following the standard procedure in the literature (Shi et al., 2020; Luo et al., 2021; Jo et al., 2022; 2023), we kekulize the molecules using the RDKit library (Landrum et al., 2006) and remove the hydrogen atoms from the molecules in the QM9 and ZINC250k datasets.

**Evaluation metrics** Following the evaluation setting of Jo et al. (2022), we generate 10,000 molecules for each dataset and evaluate them with four metrics: the ratio of valid molecules without correction (**Val.**). Frechet ChemNet Distance (**FCD**) evaluates the chemical properties of the molecules by measuring the distance between the feature vectors of generated molecules and those in the test set using ChemNet. Neighborhood Subgraph Pairwise Distance Kernel (**NSPDK**) assesses the quality of the graph structure by measuring the MMD between the generated molecular graphs and the molecular graphs from the test set. Scaffold Similarity (**Scaf.**) evaluates the ability to generate similar substructures by measuring the cosine similarity of the frequencies of Bemis-Murcko scaffolds (Bemis & Murcko, 1996).

**Implementation details** We follow the evaluation setting of Jo et al. (2022; 2023) to generate 10,000 molecules and evaluate graphs with test data for each dataset. We quote the baselines results from Jo et al. (2023). For a fair comparison, we adopt the Graph Transformer (Dwivedi & Bresson, 2020; Vignac et al., 2023) as the neural network used in GDSS+Transformer (TF) (Jo et al., 2022), DiGress (Vignac et al., 2023), and DruM (Jo et al., 2023). We apply the exponential moving average (EMA) to the parameters while sampling and set the diffusion steps to 1000 for all the diffusion models. For both QM9 and ZINC250k, we encode nodes and edges to one-hot and set $S_{cale} = 0.9$, $S_{hift} = 0.09$. For $\eta$ modulated in molecule generation, with the help of chemical knowledge, we apply $\eta = [10000, 100, 100, 100, 30]$ to carbon-carbon bonds, carbon-nitrogen, carbon-oxygen, carbon-fluorine, and other possible bonds, respectively. For $\eta$ of nodes, we apply $\eta = [10000, 100, 100, 30]$ on carbon atom, nitrogen atom, oxygen atom and other possible

Table 6: Additional 2D molecule generation results on QM9 dataset.

| | QM9 ($_{|N| \leq 9}$) | | | | | |
|---|---|---|---|---|---|---|
| **Method** | Valid (%) ↑ | FCD ↓ | NSPDK ↓ | Scaf. ↑ | Uniq (%) ↑ | Novelty (%) ↑ |
| MoFlow | 91.36 | 4.467 | 0.0169 | 0.1447 | 98.65 | **94.72** |
| GraphAF | 74.43 | 5.625 | 0.0207 | 0.3046 | 88.64 | 86.59 |
| GraphDF | 93.88 | 10.928 | 0.0636 | 0.0978 | 98.58 | 98.54 |
| EDP-GNN | 47.52 | 2.680 | 0.0046 | 0.3270 | 99.25 | 86.58 |
| GDSS | 95.72 | 2.900 | 0.0033 | 0.6983 | 98.46 | 86.27 |
| GDSS+TF | 99.68 | 0.737 | 0.0024 | 0.9129 | - | - |
| DiGress | 98.19 | 0.095 | 0.0003 | 0.9353 | 96.67 | 25.58 |
| SwinGNN | 99.71 | 0.125 | 0.0003 | - | 96.25 | 17.34 |
| GraphARM | 90.25 | 1.22 | 0.0020 | - | 95.62 | 70.39 |
| EDGE | 99.10 | 0.458 | - | 0.763 | **100.0** | - |
| GruM | 99.69 | 0.108 | **0.0002** | 0.9449 | 96.90 | 24.15 |
| **GBD** | **99.88** | **0.093** | **0.0002** | **0.9510** | 97.12 | 26.32 |

Table 7: Additional 2D molecule generation results on ZINC250k dataset.

| | ZINC250k ($_{|N| \leq 38}$) | | | | | |
|---|---|---|---|---|---|---|
| **Method** | Valid (%) ↑ | FCD ↓ | NSPDK ↓ | Scaf. ↑ | Uniq (%) ↑ | Novelty (%) ↑ |
| MoFlow | 63.11 | 20.931 | 0.0455 | 0.0133 | **99.99** | **100.0** |
| GraphAF | 68.47 | 16.023 | 0.0442 | 0.0672 | 98.64 | 99.99 |
| GraphDF | 90.61 | 33.546 | 0.1770 | 0.0000 | 99.63 | **100.0** |
| EDP-GNN | 82.97 | 16.737 | 0.0485 | 0.0000 | 99.79 | **100.0** |
| GDSS | 97.01 | 14.656 | 0.0195 | 0.0467 | 99.64 | **100.0** |
| GDSS+TF | 96.04 | 5.556 | 0.0326 | 0.3205 | - | - |
| DiGress | 94.99 | 3.482 | 0.0021 | 0.4163 | 99.97 | 99.99 |
| SwinGNN | 81.72 | 5.920 | 0.006 | - | 99.98 | 99.91 |
| GraphARM | 88.23 | 16.26 | 0.055 | - | 99.46 | **100.0** |
| GruM | **98.65** | 2.257 | **0.0015** | **0.5299** | 99.97 | 99.98 |
| **GBD** | 97.87 | **2.248** | 0.0018 | 0.5042 | 99.97 | 99.99 |

atoms, respectively. As described in Section 2.3, applying the appropriate $\eta$ for different node types and edge types can prolong the presence of related substructures during the diffusion process. In practice, we set threshold as 0.5 to quantize generated continue adjacency matrix, and the value in discrete adjacency matrix is 0 after quantizing if and only if all values in each dimension are all 0.

**Complete results on 2D molecule generation**    We provided additional results including Unique and Novelty on 2D molecule generation in Table 6 and Table 7.

### E.3 COMPUTING RESOURCES

For all experiments, we utilized the PyTorch (Paszke et al., 2019) framework to implement GBD and trained the model with NVIDIA GeForce RTX 4090 and RTX A5000 GPUs.

## F ADDITIONAL EXPERIMENTAL RESULTS

### F.1 EVALUATION WITH LARGER SAMPLE SIZE

As described in Section 4.1, small number of nodes and the insufficient size of the sampled graphs can lead to large standard deviations when evaluating with the reference graph on smaller dataset. Therefore, we attempted to evaluate the large number of generated graphs and report the results in Table 8. We observe that our proposed GBD outperforms previous continuous and discrete diffusion models on both smaller datasets. Furthermore, GBD significantly surpasses the wavelet-based diffusion model (Wave-GD) by a wide margin on the Community-small dataset, as evidenced by both means and standard deviations, Specifically, GBD achieves 85.0%, 90.5%, and 40.0% improvements over Wave-GD in the MMDs means of Degree, Cluster, and Orbit, respectively, indicating that our

proposed model is capable of generating smaller graphs that are closer to the data distribution with better stability.

Table 8: Generic graph generation results with enlarged sample (1024 graphs).

| Method | Ego-small | | | | Community-small | | | |
|---|---|---|---|---|---|---|---|---|
| | Deg. ↓ | Clus. ↓ | Orbit. ↓ | Avg. ↓ | Deg. ↓ | Clus. ↓ | Orbit. ↓ | Avg. ↓ |
| GraphRNN | 0.040 | 0.050 | 0.060 | 0.050 | 0.030 | 0.010 | 0.010 | 0.017 |
| GNF | 0.010 | 0.030 | 0.001 | 0.014 | 0.120 | 0.150 | 0.020 | 0.097 |
| EDP-GNN | 0.010 | 0.025 | 0.003 | 0.013 | 0.006 | 0.127 | 0.018 | 0.050 |
| GDSS | 0.023 | 0.020 | 0.005 | 0.016 | 0.029 | 0.068 | 0.004 | 0.034 |
| ConGress* | 0.030 ($\pm$ 0.001) | 0.050 ($\pm$ 0.003) | 0.008 ($\pm$ 0.001) | 0.030 - | 0.004 ($\pm$ 0.000) | 0.047 ($\pm$ 0.000) | **0.001** ($\pm$ 0.000) | 0.017 - |
| DiGress* | 0.009 ($\pm$ 0.000) | 0.031 ($\pm$ 0.002) | **0.003** ($\pm$ 0.000) | 0.014 - | 0.003 ($\pm$ 0.000) | 0.009 ($\pm$ 0.001) | **0.001** ($\pm$ 0.000) | 0.004 - |
| Wave-GD | 0.010 ($\pm$ 0.001) | 0.018 ($\pm$ 0.003) | 0.005 ($\pm$ 0.002) | 0.011 - | 0.016 ($\pm$ 0.000) | 0.077 ($\pm$ 0.006) | **0.001** ($\pm$ 0.002) | 0.031 - |
| **GBD** | **0.007** ($\pm$ 0.000) | **0.011** ($\pm$ 0.001) | **0.003** ($\pm$ 0.000) | **0.010** - | **0.002** ($\pm$ 0.000) | **0.007** ($\pm$ 0.001) | **0.001** ($\pm$ 0.000) | **0.003** - |

## F.2 COMPARISON ON VARIOUS NODE FEATURE INITIALIZATION.

**Node feature initialization** We initialize node representations using the following node-level features, respectively:

- **Degree (one-hot)**: Degree with one-hot format is a categorical representation of a node's degree. It encodes the degree information as a binary vector where each position corresponds to a possible degree value. The position corresponding to the node's actual degree is set to 1, while all other positions are 0.

- **Degree (normalized)**: The normalized degree is a continuous representation of a node's degree, scaled to a value between 0 and 1.

- **Centrality**: Here we adopt the normalized betweenness centrality as initial node features and it is a measure of a node's importance based on its role in connecting other nodes. It quantifies the fraction of shortest paths between all pairs of nodes that pass through the given node. The normalized version scales this value to be between 0 and 1.

- **Eigenvectors**: Following Jo et al. (2022), we adopt the two first eigenvectors associated to non zero eigenvalues as initial node features.

**Results** As shown in Table 9, GBD outperforms GDSS + TF and ConGress by a large margin in all MMDs when the node representation exhibits sparsity and long-tailedness. Additionally, GBD achieves competitive performance compared with other Gaussian-based diffusion models while the node feature is initializing with Eigenvectors. This demonstrates that our proposed GBD has the ability to model graphs with flexible node features, indicating its potential for modeling graphs with more informative features.

## G VISUALIZATION

We follow the implementation described in Section 2.3 and the nodes in all adjacency matrices are reordered by decreasing the degree of nodes. Apparently, we can find that edges associated with nodes with large degree will be the first to be identified and then spread in decreasing order of degree on both datasets in Appendix G.1. It is worth noting that the reverse beta diffusion can converge rapidly, leading to generated graphs with correct topology at an early stage. This shows that our proposed GBD can further explore the potential benefits of beta diffusion, resulting in valid graphs with stability and high quality. For generated molecule graphs shown in Appendix G.1, we can observe that GBD can successfully generate valid and high-quality 2D molecules, verifying its ability to model attributed graphs. More generated graphs are presented following.

Table 9: The effect of Feature Initialization on `Community-small` and `Ego-small`.

| | Community-samll | | | | | | | | | | | |
|---|---|---|---|---|---|---|---|---|---|---|---|---|
| **Node Feature** | Degree (one-hot) | | | Degree (normalized) | | | Centralities | | | Eigenvectors | | |
| **Method** | Deg. | Clus. | Orbit. | Deg. | Clus. | Orbit. | Deg. | Clus. | Orbit. | Deg. | Clus. | Orbit. |
| GDSS+TF | 0.008 | 0.080 | 0.005 | 0.009 | 0.077 | 0.005 | 0.010 | 0.075 | 0.004 | 0.005 | 0.061 | 0.003 |
| ConGress | 0.024 | 0.072 | 0.006 | 0.020 | 0.076 | 0.006 | 0.013 | 0.079 | 0.005 | 0.004 | 0.067 | 0.003 |
| GBD | 0.002 | 0.060 | 0.002 | 0.004 | 0.059 | 0.003 | 0.003 | 0.059 | 0.003 | 0.004 | 0.064 | 0.002 |

| | Ego-samll | | | | | | | | | | | |
|---|---|---|---|---|---|---|---|---|---|---|---|---|
| **Node Feature** | Degree (one-hot) | | | Degree (normlized) | | | Centralities | | | Eigenvectors | | |
| **Method** | Deg. | Clus. | Orbit. | Deg. | Clus. | Orbit. | Deg. | Clus. | Orbit. | Deg. | Clus. | Orbit. |
| GDSS+TF | 0.016 | 0.020 | 0.004 | 0.018 | 0.023 | 0.005 | 0.017 | 0.026 | 0.006 | 0.013 | 0.015 | 0.002 |
| ConGress | 0.045 | 0.059 | 0.015 | 0.037 | 0.064 | 0.017 | 0.032 | 0.057 | 0.014 | 0.020 | 0.029 | 0.011 |
| GBD | 0.011 | 0.014 | 0.002 | 0.013 | 0.017 | 0.002 | 0.015 | 0.012 | 0.003 | 0.017 | 0.016 | 0.002 |

## G.1 GENERATIVE PROCESS OF GBD ON GENERAL DATASETS

We visualize the generative process of GBD on the `Community-small` and the `Ego-small` dataset in Figures 5 and 6, respectively.

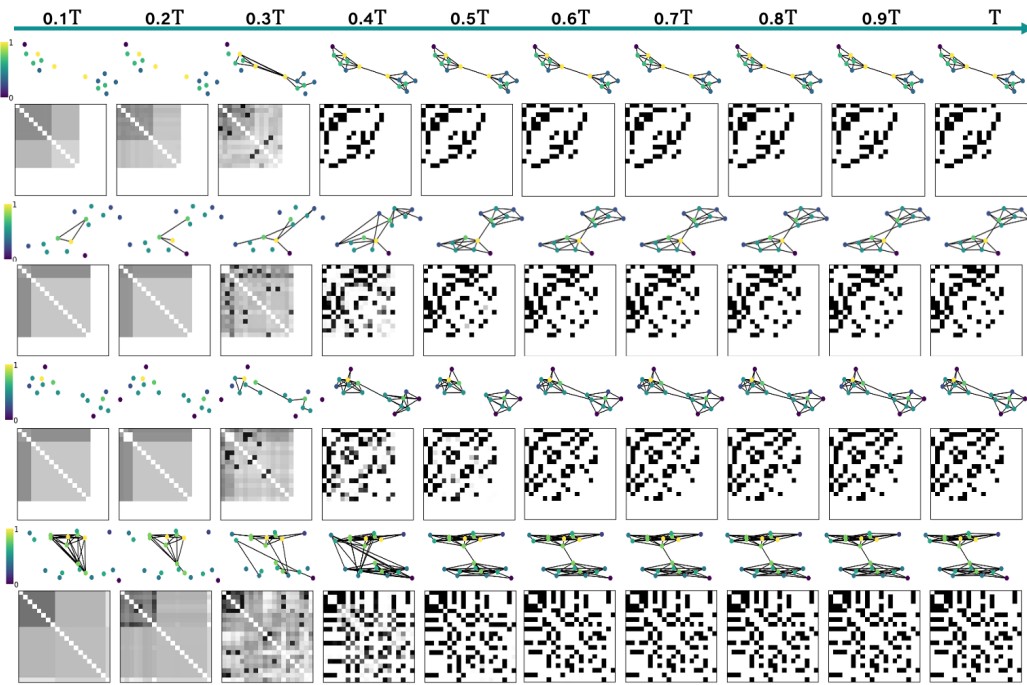

Figure 5: Visualization of the generative process of GBD on the `Community-small` dataset.

## G.2 GENERATIVE PROCESS OF GBD ON COMPLEX GRAPH

We provide the visualization of the complex graph generated by GBD on the `Planar` and the `SBM` datasets in Figure 7 and in Figure 8, respectively.

## G.3 GENERATED GRAPHS OF GBD ON 2D MOLECULE DATASETS

We provide the visualization of the 2D molecules generated by GBD on the `QM9` and the `ZINC250k` datasets in Figure 9 and in Figure 10, respectively.

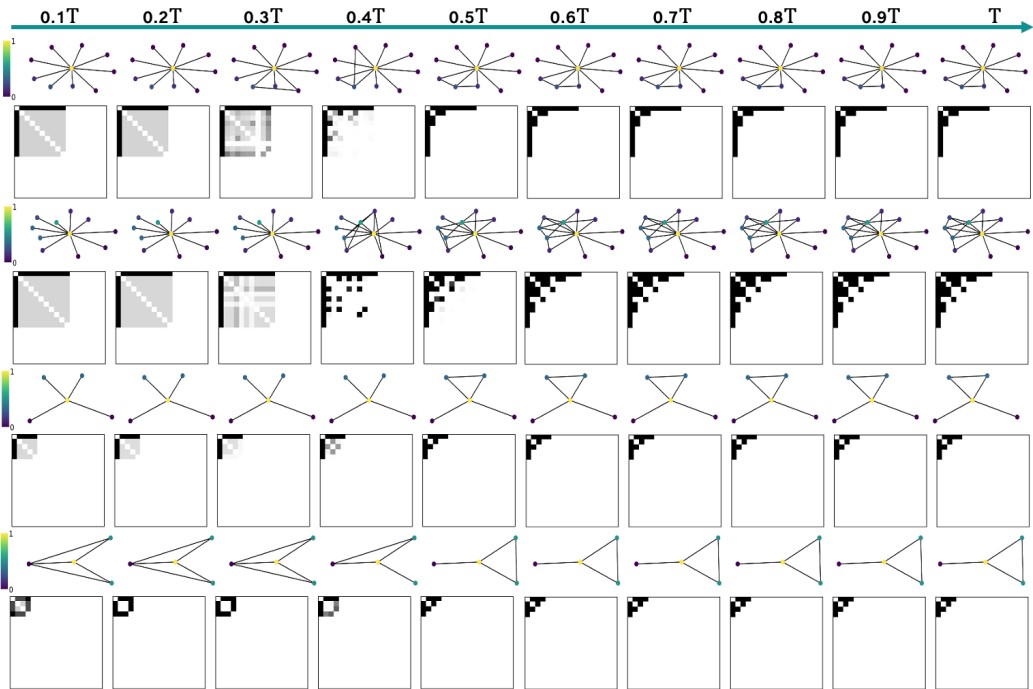

Figure 6: Visualization of the generative process of GBD on the `Ego-small` dataset.

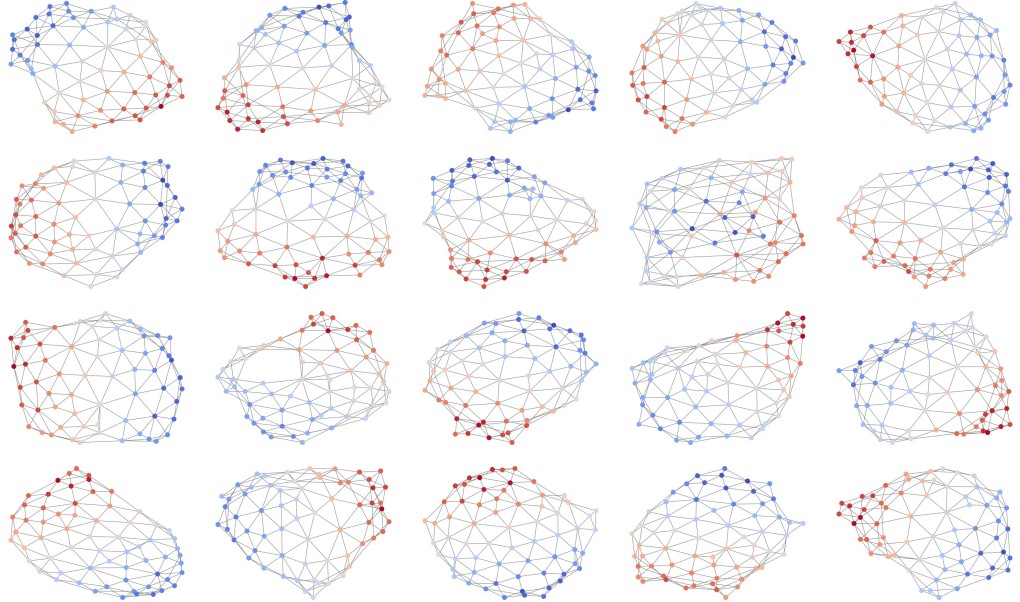

Figure 7: Visualization of the generated graphs of GBD on the `Planar` dataset.

## G.4 GENERATED GRAPHS OF GBD ON BA-NETWORKS

We provide the visualization of the BA-networks ($n = 20, m = 2$) generated by GBD in Figure 11.

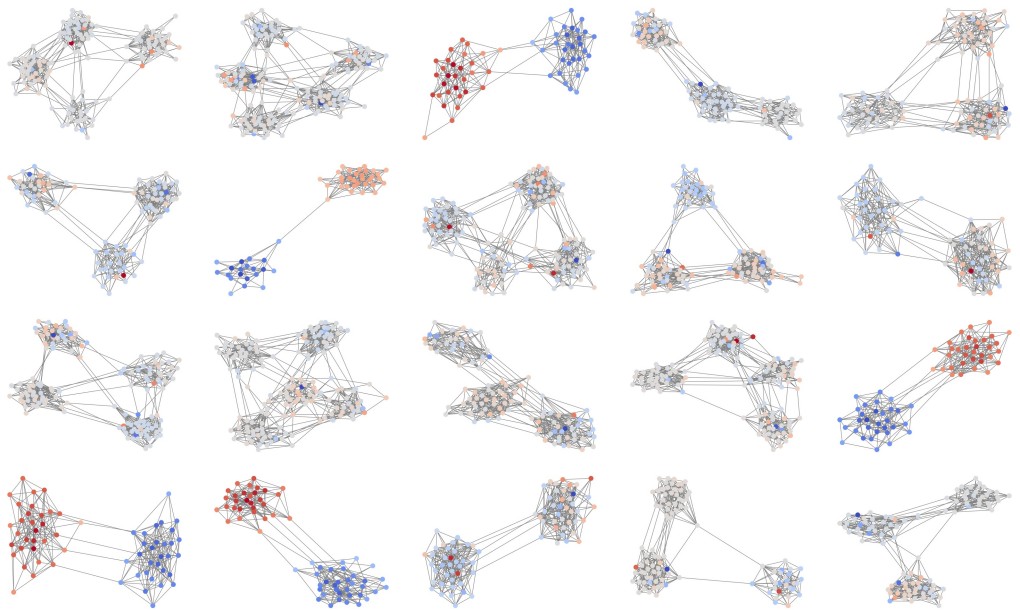

Figure 8: Visualization of the generated graphs of GBD on the SBM dataset.

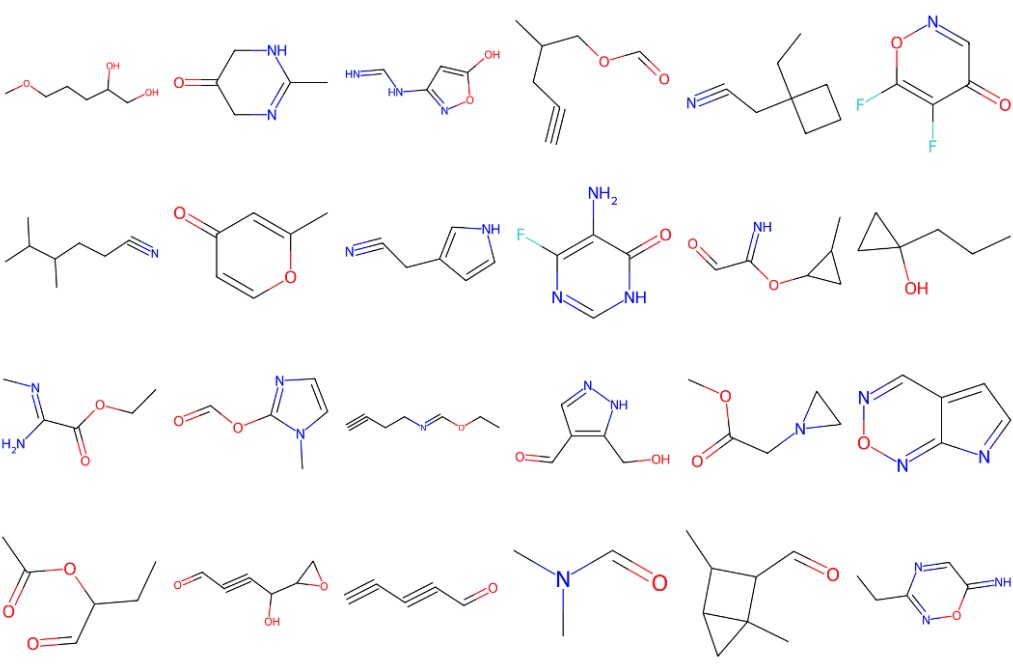

Figure 9: Visualization of the generated graphs of GBD on the QM9 dataset.

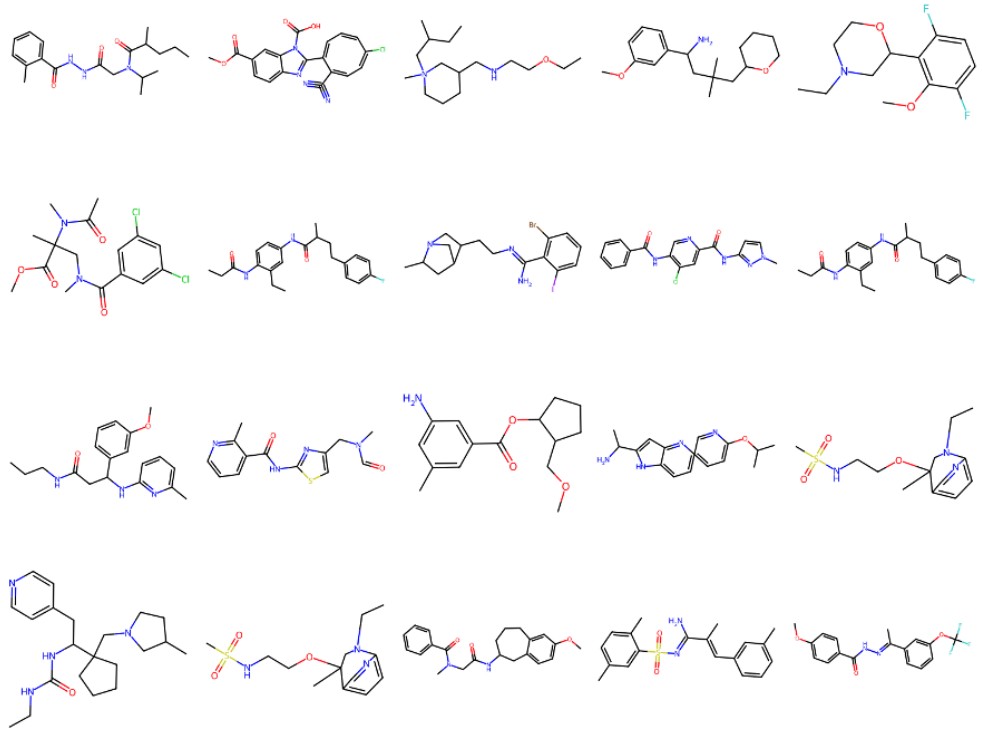

Figure 10: Visualization of the generative graphs of GBD on the `ZINC250k` dataset.

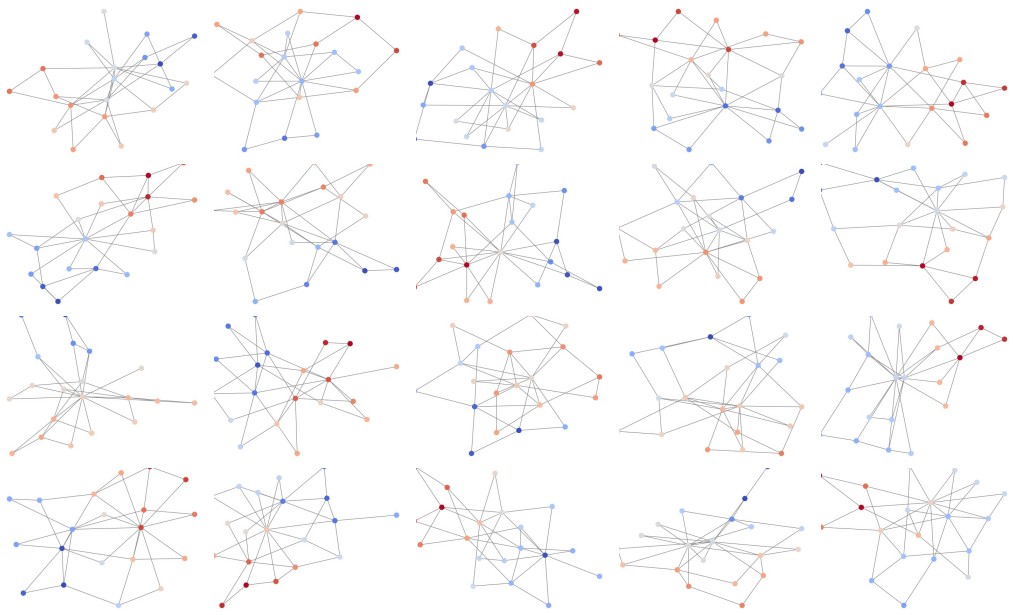

Figure 11: Visualization of the generative graphs of GBD on the `BA-network` ($n = 20, m = 2$).

