# OpenReview forum: "Advancing Graph Generation through Beta Diffusion"
_ICLR.cc/2025/Conference — ICLR 2025 Poster_

### Official Review · Reviewer_PnYi · 2024-10-29

**Soundness:** 3
**Presentation:** 4
**Contribution:** 3
**Rating:** 6
**Confidence:** 5

**Summary:**

This paper presents a Graph Beta Diffusion for the natural conflict between graph structure and diffusion model. A modulation technique is proposed to stabilize the topology of the key graph.

**Strengths:**

1. The motivation of this paper is very reasonable. Diffusion model and graph structure are incompatible.
2. The paper is well written and is easy to understand.
3. The Concentration modulation method is very cleverly designed. It seems to make good use of the topological properties of the graph structure.

**Weaknesses:**

1. The authors give no clear evidence as to why beta diffusion is effective at modeling the graph structure. The discrete distribution can take many forms. Or why not use BFN to model the distribution of a graph.
2. In terms of experimental results, if concentration modulation is not included, the effect of beta diffusion is not better than Digress and Grum. This does not effectively explain the effectiveness of beta diffusion. Another point of view is, can the ordinary diffusion model also use this idea to control the noise process?

**Questions:**

1. How to explain the beta distribution is better suited to the graph than other discrete distributions.
2. Can concentration modulation be used with other diffusion methods?

---

> ### Author Response · Authors · 2024-11-21
>
> We sincerely appreciate your valuable feedback. We are pleased that you found it logical to apply a beta distribution-based diffusion method to graph modeling and recognized the novelty of our concentration modulation technique! In response to the concerns you raised, we provide detailed answers below:
>
> 1. **The authors give no clear evidence as to why beta diffusion is effective at modeling the graph structure. The discrete distribution can take many forms. Or why not use BFN to model the distribution of a graph.**
>
> Regarding the motivation behind choosing the beta diffusion model over other discrete models, please refer to the second part of our discussion in the “global response”. We are not aware of the application of BFN to graph generations in existing literature and appreciate the reviewer for providing more guidance on this point.
>
> 2. **In terms of experimental results, if concentration modulation is not included, the effect of beta diffusion is not better than Digress and GruM.**
>
> We would like to clarify that, as demonstrated by the results in Tables 2, 3, and 5, our model has already achieved clear improvement in most cases compared to DiGres, even without concentration modulation, particularly in molecule generation. This highlights the power and potential of modeling graph data using our proposed Graph Beta Diffusion (GBD) model.
>
> To further adapt the proposed model to graph generation tasks, we developed a concentration modulation technique that incorporates inductive biases based on the statistical properties of graphs. This novel technique further boosts the performance of GBD across various graph generation tasks. These results not only validate the effectiveness of our concentration modulation technique but also inspire future exploration of Beta diffusion modeling in broader scenarios.
>
> 3. **Can the ordinary diffusion model also use this idea to control the noise process?**
>
> This is an excellent question, and addressing it will help clarify the unique advantages of beta diffusion in graph modeling. To begin, please refer to the first point in our Global Response. Below, we provide additional discussion.
>
> From the perspective of our motivation, the concentration modulation technique in our model plays a crucial role in controlling the mixing process, as described in Section 2.3. Since the initial state of the reverse beta diffusion process is set to 0, important graph structures emerge first (state > 0) under an appropriately chosen concentration parameter. This allows us to predict the topology more accurately and model the graph more efficiently. In contrast, with an uncertain initial state (usually sampled from $N(0, I)$), it remains an open question how Gaussian diffusion processes could effectively describe the concept of "first appearance of important structures." This gap motivates us to continue exploring this technique in future research.
>
> While the concentration modulation technique enhances performance, we emphasize that the successful adaptation of beta diffusion to graph modeling is primarily due to the synergy between the inherent flexibility of beta diffusion and the meticulous design of our GBD framework.

---

> > ### Comment · Reviewer_PnYi · 2024-11-22
> >
> > Thanks for your response. I will keep my positive rating.

---

> > > ### Author Response · Authors · 2024-11-28
> > >
> > > We thank reviewer for recognizing our work!

---

### Official Review · Reviewer_9sdC · 2024-10-31

**Soundness:** 3
**Presentation:** 3
**Contribution:** 4
**Rating:** 6
**Confidence:** 4

**Summary:**

This paper introduces Graph Beta Diffusion to address the inherent conflict between graph structure and diffusion models. It also proposes a modulation technique aimed at stabilizing the topology of the key graph.

**Strengths:**

1.The paper is well-articulated and easy to comprehend.
2.The rationale behind this paper is quite sound, as diffusion models and graph structures do not align well.

**Weaknesses:**

1.Could the author offer a comparison of complexity? For instance, the calculation of concentration appears to be time-consuming. Authors should provide a comparison of complexity with other methods, preferably including both theoretical analysis as well as experimental data. (e.g. complexity comparison between GDSS and Digress)
2.Lack of detection for networks with other topological properties. BA networks, for example, follow a power-law distribution.
3.The sample rate analysis and scalability of this method are not discussed in detail. For example, in an SDE-based diffusion framework, we can use many different accelerated sampling methods, as well as perform conditional generation. But is this approach effective?

**Questions:**

1.For a graphset data, a concentration modulation is calculated for each graph and then concat them into a matrix. Will this cause a large consumption of time? And how should Concentration modulation be defined during generation? Because we don't know some of the properties of the graph in advance when sampling from the beta distribution?
2.What is the sampling rate of this method? And how does the sampling rate affect the quality of the generated graphs?
3.Can this approach be extended to conditional diffusion generation? Other methods like GDSS or Digress can easily combine with conditional generation in theory.

**Details Of Ethics Concerns:**

No.

---

> ### Author Response · Authors · 2024-11-21
>
> We sincerely appreciate your valuable feedback and thank you for highlighting the contribution of our work!
>
> In response to the concerns you raised, we provide detailed answers below:
>
> 1. **Details about complexity.**
>
> Following the design of Graph Transformer in GruM and DiGree, the overall memory and time complexity of our network is O(n^2) each layer. In addition, we set up the same network configuration in the specific experiments as in DiGress.
>
> Regarding the detailed elucidation about concentration modulation during the sampling stage, please refer to the first part of our discussion in the "global response" and we note that the cost of applying the summary statistics obtained from the training data during sampling is negligible.
>
> 2. **Lack of detection for networks with other topological properties. BA networks, for example, follow a power-law distribution.**
>
> Generally, the BA model primarily focuses on power-law distribution, like degree distribution, and may not capture other important features or attributes.
> It is worth noting that graph data elements exhibit diverse probabilistic characteristics and it is the most important motivation to adapt beta diffusion to graph modeling, as detailed in the second part of our discussion in the “global response”.
> Therefore, we model these complex data with advanced diffusion models, providing powerful ability to model the joint distribution of various types of elements within graph data.
>
> 3. **Details about the sample rate analysis and scalability of this method.**
>
> Follow your suggestion, we add the ablation of different sampling rates on Planar as below. From the results in this Table, We can find that the performance of all methods decreases when the number of sampling steps is reduced to 300, but our model is still able to generate high-quality graphs with 75% V.U.N. and have the best V.U.N. when the number of sample steps is greater than 500.
>
> **V.U.N (%) on Planar**
>
> | sample steps | DiGress | GruM | **GBD** |
> |-----|---------|------|---------|
> | 300          | 65.0    | **75.0** | **75.0**    |
> | 500          | 75.0    | 87.5 | **90.0**    |
> | 700          | 80.0    | **90.0** | **90.0**    |
> | 1000         | 75.0    | 90.0 | **92.5**    |
>
> 4. **Clarification about graph conditional generation.**
>
> Thanks for your insightful question about graph conditional generation! For the graph conditional generation task, we note that the neural network and generative paradigm are the same as in DiGress, so our model can also have the same ability to generate graphs with conditional properties like DiGress does - incorporating the label or properties into the network as a global feature directly.
>
> However, since the focus of this paper is on unconditional graph generation and providing a comprehensive comparison with existing methods, this topic lies beyond the scope of our current work. Moreover, effective conditional graph generation often requires more than simply injecting label conditions; it typically involves various preference optimization techniques, which are distinct research areas worth exploring in their own right.

---

> > ### Comment · Reviewer_9sdC · 2024-11-22
> > **Response to  Rebottal**
> >
> > Thanks for your response! Most of my concern has been resolved. However, I would still like the authors to add experiments on the BA network. Due to the nature of its power law distribution, it is more reflective of the effect of your node concentration modulation. Therefore, I think this experiment is very helpful for you to illustrate the advantages of the model.

---

> ### Author Response · Authors · 2024-11-24
>
> Thanks for your replies and valuable suggestion about experiments on the BA networks!
>
> Follow your suggestion, we conduct additional experiments and show the effects of our proposed concentration modulation on BA networks. Specifically, we construct two groups of BA-networks with where $n=[20, 100]$ and corresponding $m=[2, 3]$, respectively. $n$ denotes the total number of node per graphs and $m$ denotes the number of edges that each new node will attach to existing nodes. In the tables below, we compare the results of our proposed model and its vanilla vision which denoted as $GBD$ and $GBD_{w/o -cm}$, respectively. The results from these tables demonstrate that our proposed node concentration modulation can provide positive effects on BA networks where the node degrees follow distinct power-law distributions. We will provide details in the revision.
>
> Additionally, we have provided the corresponding visualization of BA-networks ($n=20, m=2$) generated with our model in the revision, please refer to Appendix F.4.
>
>
> **Results of (n=20, m=2)**
> |             | Degree     | Clustering | Orbit     | Spectral  |
> |---|--|---|-----|----|
> | Training Set| 0.0011     | 0.0140     | 0.0008    | 0.0034    |
> | $GBD_{w/o-cm}$     | 0.0019  | 0.019  | 0.0021 | 0.0063 |
> | $GBD$     | **0.0014** | **0.018**  | **0.0010**  | **0.0054**  |
>
> **Results of (n=100, m=3)**
> |             | Degree     | Clustering | Orbit     | Spectral  |
> |---|--|---|-----|----|
> | Training Set| 0.0004     | 0.0134     | 0.0086    | 0.0021    |
> | $GBD_{w/o-cm}$     | 0.0010  | 0.0195  | **0.0445** | 0.0020 |
> | $GBD$     | **0.0008** | **0.0162**  | 0.0469  | **0.0012**  |

---

> > ### Comment · Reviewer_9sdC · 2024-11-25
> > **Response**
> >
> > Thanks for your response! My concern has been solved now. I will raise my score.

---

> > > ### Author Response · Authors · 2024-11-28
> > >
> > > We thank reviewer for raising the initial ratings to the acceptance line. We are happy to see our rebuttal address your concerns.

---

### Official Review · Reviewer_ioTm · 2024-11-03

**Soundness:** 3
**Presentation:** 3
**Contribution:** 2
**Rating:** 6
**Confidence:** 3

**Summary:**

This paper presents a novel graph generation model by adapting the recent diffusion process work (Zhou et al., 2023) to handle graph-structured data. The authors assess the performance of the proposed model against state-of-the-art methods on both synthetic and real-world datasets, including molecular datasets commonly used in graph generation tasks.

**Strengths:**

- Adapting the recent beta diffusion process to graph-structured data.
- The model is evaluated on both synthetic and real-world data, including widely used molecule datasets.
- The structure of the paper is generally well-organized.

**Weaknesses:**

- While the model adapts the diffusion process for graph data, there is limited discussion on unique contributions beyond this adaptation. Emphasizing the model’s distinct aspects and theoretical advancements would strengthen the paper's impact.
- Similarly, the paper heavily references Zhou et al. (2023), making it challenging for readers who are not familiar with this work to fully understand the technical content.
- The reported results raise questions about the practical significance of the proposed architecture, with inconsistent comparisons across tables (missing baselines and metrics like Spec. and V.U.N.), lack of standard deviation reporting in Tables 2 and 3, and missing entries, all of which reduce confidence in the experimental evaluation.

**Questions:**

**Questions:**

- The methodology mainly adapts the work of Zhou et al. (2023) for graph-structured data and raises concerns about the originality of the paper. Could the authors emphasize their unique contributions beyond this adaptation?
- The reported improvements over baseline methods appear minimal. Could the authors clarify the practical impact of these results?
- Tables 1 and 2 use different baseline models, omitting certain comparisons (e.g., Wave-GD, GraphARM, GNF, GraphVAE in Table 1; GruM, GDSS+TF, SPECTRE in Table 2). Tables are missing standard deviation values (Tables 2 and 3) and certain metrics (e.g., Spec. and V.U.N in Table 1), with no explanation for missing entries. Could the authors clarify their choices?
- Does the proposed model offer any guarantees regarding the connectivity of the generated graphs?


**Additional comments:**
- The provided equations in the paper are a bit challenging to follow, especially for readers unfamiliar with the work of Zhou et al. (2023). The authors might provide additional background or a summary of this prior work to improve readability.
- The graph images in the first row of Figure 2 are unclear, and it is difficult to see the nodes and edges. The authors might consider improving the quality or resolution of these visuals.
- Similarly, the SBM graph in Figure 3 is hard to interpret due to its size. Providing an adjacency matrix or alternative visualization might enhance readability.

Given the concerns outlined above regarding the experimental evaluations and originality of the paper, I recommend a rating of 5 for the paper.

---

> ### Author Response · Authors · 2024-11-21
>
> Thank you for your thoughtful review and for recognizing the strengths of our work. We are glad to hear that you valued our adaptation of the beta diffusion process to graph-structured data, the diverse evaluation on both synthetic and real-world datasets (including molecular benchmarks), and the clear organization of the paper.
>
> In response to the concerns you raised, we provide detailed answers below:
>
> 1. **Regarding the unique contribution for our proposed model:**
>
> Beyond the adaptation of beta diffusion to graph modeling, we note that several designs have been proposed to improve the performance of modeling graph data in Section 2.3, especially the concentration modulation technique we further elucidated in the first part of our discussion of “global response”.
> With this novel technique, we are able to improve the performance of GBD in a variety of graph generation tasks, which not only shows the effectiveness of our concentration modulation technique, but also brings inspiration to utilize beta diffusion model in more scenarios in the future.
>
> 2. **Insufficient background of beta diffusion**
>
> Thanks for your valuable suggestion. Following your advice, we will add more technical details about Zhou et al. (2023) in the revision.
>
> 3. **Regarding the questions about results of experiments, we clarify as follows:**
>
> We sincerely appreciate your valuable feedback and careful reading!
>
> For the missing baselines, we note that some state-of-art baselines do not include all the main experiments we have done in their own paper. To make the comparison as comprehensive and fair as possible, we used a wider range of baselines and did more experiments in our submission, demonstrating the strong modeling ability of our model in graph generation tasks.
>
> For the missing metrics of Spec. and V.U.N., we note that measuring the performance of a model with these metrics on **simple and small graphs** is not an adequate reflection of the models’ modeling capabilities
> Therefore, to better measure the ability of modeling real-data for graph generative models, it is necessary to evaluate these metrics on graphs with complex structure and large scale.
>
> With these metrics reported in Table 2 for complex graphs, as well as the valid, unique, and novelty for molecule graphs in Appendix C.2, we believe our proposed model is capable of modeling complex graphs and achieves great performance on various datasets.
>
> For the miss standard deviation, we have reported in the following tables and will update Table 2 and Table 3 in the revision.
>
> **Planar**
>
> |       | Deg.   | Clus.  | Orbit, | Sepc.  | V.U.N. (%) |
> |----|---|------|--------|--------|--------|
> | $\mu$    | 0.0003 | 0.0353 | 0.0135 | 0.0059 | 92.5   |
> | $\sigma$ | 0.0000 | 0.0009 | 0.0017 | 0.0004 | 5.4    |
>
> **SBM**
>
> |       | Deg.   | Clus.  | Orbit, | Sepc.  | V.U.N. (%) |
> |--|----|-|--------|--------|--------|
> | $\mu$    | 0.0013 | 0.0493 | 0.0446 | 0.0047 | 75.0   |
> | $\sigma$ | 0.0001 | 0.0023 | 0.0003 | 0.0003 | 4.0    |
>
> **QM9**
>
> |       | Valid (%)  | FCD.   | NSPDK  | Scaf.  |
> |-------|--------|--------|--------|--------|
> | $\mu$   | 99.88  | 0.093  | 0.0002 | 0.9510 |
> | $\sigma$ | 0.06   | 0.0023 | 0.0000 | 0.0057 |
>
> **ZINC250k**
>
> |       | Valid (%) | FCD.  | NSPDK  | Scaf.  |
> |-------|--|---|---|--------|
> | $\mu$    | 97.87 | 2.248 | 0.0018 | 0.5042 |
> | $\sigma$ | 0.21  | 0.006 | 0.0005 | 0.0100 |
>
> 4. **Regarding the marginal improvements over baselines in some cases:**
>
> Thanks for your careful reading! We note that our proposed GBD has significant improvement on all experiments compared to either continuous or discrete diffusion models (GDSS+TF, ConGress and DiGress), demonstrating the GBD’s comprehensive ability of graph modeling. Compared to the state-of-art baselines (GraphARM, SwinGNN, EDGE and GruM), our GBD outperforms these baselines in most cases, and achieves competitive performance compared to GruM. Additionally, we also provide rich visualizations in Appendix F, showing that our GBD can generate high-quality valid graphs both on complex general and molecular graphs.
>
> 5. **Regarding the connectivity guarantee while modeling graphs.**
>
> Our model does not provide the explicit guarantees for modeling the connectivity of graphs, so do other baselines.
>
> Following your advice, we report the connectivity metric evaluated on Planar and SBM as below. From the results in the Table below, we can find that our model can always generate connected graphs for both two complex graphs, demonstrating that our model has the ability to implicitly constrain connectivity very well.
>
> **Connectivity**
>
> |      | Planar | SBM   |
> |------|--|-------|
> | **GBD**  | 1.0    | 0.995 |
>
>
> 6. **Additional comments.**
>
> Thanks for your suggestive comments!
> We will provide more additional background about Zhou et al. (2023) in the revision and improve the visualization, including improving the quality of Figure 2 and adding the additional adjacency matrix of SBM in the revision.

---

> ### Comment · Area_Chair_LinD · 2024-11-25
>
> Could please acknowledge and respond to the rebuttal.

---

> > ### Comment · Reviewer_ioTm · 2024-11-25
> > **Thank you!**
> >
> > I thank the authors for addressing my questions, and I have raised my rating.

---

> > > ### Author Response · Authors · 2024-11-28
> > >
> > > We are glad that our response addressed your concerns. We thank the reviewer for raising the initial ratings to the acceptance level.

---

### Official Review · Reviewer_gBaw · 2024-11-04

**Soundness:** 3
**Presentation:** 2
**Contribution:** 3
**Rating:** 6
**Confidence:** 2

**Summary:**

The paper proposes graph beta diffusion for the task of generating small graphs such as molecules. The core concept is approaching this task by adapting beta diffusion (Zhou et al., 2023), which is meant for generating data within bounded ranges (e.g., 0 to 1 for non-edge and edge) and is based on the beta diffusion rather than the Gaussian distribution. In addition to this core idea, the paper also explores several design ideas to strengthen the proposed method, including "concentration modulation," which modify the diffusion distribution for important positions in the graph. The paper concludes with experiments on real and synthetic datasets, showing that the proposed method is at least competitive with other recently proposed methods in terms of matching certain graph statistics. There are also ablation experiments supporting the lift from the design ideas.

**Strengths:**

- The core idea of applying a diffusion method based on the beta distribution to graph generation seems logical.
- The graph generation task that this paper tackles is enjoying a lot of attention recently.
- The writing is generally clear.
- The evaluation is fairly extensive, showing that the method is at least competitive with other recent approaches, and also includes ablations for the proposed design components.

**Weaknesses:**

- Since the main thrust of this paper is approaching graph generation with a diffusion method that is suited for bounded data, like the probability of each edge, there could be more discussion of alternatives to beta diffusion. What makes beta diffusion (Zhou et al., 2023) more suited for this task as opposed to, e.g., Dirichlet Diffusion Score Model (Avdeyev et al., 2023) or Dirichlet Flow Matching (Stark et al., 2024)?
- Relatedly, given that the main thrust is application of beta diffusion, there could be more intuition given for beta diffusion itself, given that it was proposed in a recent paper.
- With the exception of concentration modulation, little of the design seems specific to graphs, making this largely an application of a prior work to a specific domain. There could be more elucidation of what is gained from using beta diffusion vs standard diffusion in a graph context, e.g., some empirical or theoretical work showing that the beta distribution is more suited to capturing certain graph motifs.
- $\omega$ seems to be a key hyperparameter controlling the weight of two losses in beta diffusion. It seems unexplored here beyond reusing the value in the beta diffusion paper.

**Questions:**

Some questions are given above. Additionally:

- As stated on line 458, this method predicts $\mathbb{E}[\mathbf{G}_0 | \mathbf{G}_t$], which is not the standard diffusion setup, and as stated on line 160, this requires a neural network that predicts the conditional expectation of $\mathbf{G}_0$ given $\mathbf{G}_t$. Could the authors please expand on this design decision and its possible consequences? In particular, is it possible this reduces the diversity of samples?
- Perhaps relatedly, regarding concentration modulation: How is $\eta$, which corresponds to important positions in the sample, set at sampling time? Is the graph distribution generated by this method equivariant to permutations of the nodes?

### Typos
- Line 376 "Transofrmer"
- Line 522 "Centrailities"

---

> ### Author Response · Authors · 2024-11-21
>
> We sincerely thank you for your thoughtful and detailed feedback. We are pleased that you found the application of a beta distribution-based diffusion method to graph generation logical, recognized the relevance of our research to current trends, and appreciated both the clarity of our writing and the comprehensive evaluation, including ablations.
>
> 1. **Regarding your question about alternatives to beta diffusion and what makes beta diffusion more suitable for graph generation:**
>
> Thank you for highlighting Dirichlet Diffusion Score Model (Avdeyev et al., 2023) and Dirichlet Flow Matching (Stark et al., 2024). We note that these models primarily focus on generative modeling of categorical data and their applications on biological sequence design. While related, our research aims extend beyond categorical variables. We are developing a diffusion model that captures the joint distribution of both categorical and numerical elements within graph data, which is not the focus of the aforementioned models.
>
> In pursuit of a thorough comparative analysis, we reviewed all papers referencing these models and found only one closely related to our work:
>
> - *Floor Eijkelboom, Grigory Bartosh, Christian Andersson Naesseth, Max Welling, and Jan-Willem van de Meent. "Variational Flow Matching for Graph Generation." arXiv preprint arXiv:2406.04843 (2024).*
>
> In their study concurrent to ours, Eijkelboom et al. (2024) introduced CatFlow, a flow matching method tailored for categorical data, and evaluated it across two abstract graph generation tasks and two molecular generation tasks. They noted: "Dirichlet Flows have not been evaluated on graph generation tasks. Our preliminary experiments, conducted using the released source code, did not yield satisfactory performance out of the box, and we decided against investing substantial time in optimizing architecture and hyperparameters."
>
> Interestingly, while our study encompasses a broader array of datasets, Eijkelboom et al. (2024) evaluated their method on four datasets that we also used. The results we achieved on these shared datasets, as detailed in the following tables, demonstrate superior performance, effectively underscoring the efficacy of our approach in graph generation tasks.
>
> | | **Ego-small**| | | **Community-small**| | |
> |-|-|-|-|-|-|-|
> | | Degree | Clustering | Orbit | Degree | Clustering | Orbit |
> | CatFlow| 0.013  | 0.024 | 0.008 |0.018  | 0.086 | 0.007 |
> | **GBD**| 0.011 | 0.014  | 0.002 |0.002 | 0.060  | 0.002 |
>
> |    | **QM9**  |  | |  **ZINC250k**|   | |   |
> |---|--|--|---|--|--|--|--|
> |        | Valid (%) | Unique | FCD | Valid (%) | Unique | FCD |
> | CatFlow| 99.81     | 99.95  | 0.441 |  99.21     | 100.00 | 12.211 |
> | **GBD**| 99.88 | 97.12  | 0.093 | 97.87 | 99.97  | 2.248  |
>
> 2. **We appreciate the opportunity to provide further insight into the beta diffusion approach, particularly as it represents a recent advancement in the field.**
>
> Traditional diffusion and flow-matching methods are generally designed to handle either continuous or discrete data types exclusively. In contrast, beta diffusion utilizes scaled and shifted beta distributions governed by multiplicative diffusion processes. This approach is adept at modeling a broad spectrum of continuous distributions and can effectively approximate discrete distributions as well. By leveraging these unique properties, beta diffusion establishes a robust and adaptable framework that is ideally suited for the diverse data types encountered in graph generation.
>
> 3. **The reviewer is hoping to see more elucidation of what is gained from using beta diffusion vs standard diffusion for graphs.**
>
> Thank you for this insightful question. The advantage of using beta diffusion in the graph context lies in its ability to model both discrete and continuous components.
>
> Unlike standard diffusion models based on Gaussian or categorical distributions that specialize in either continuous or discrete domains, beta diffusion bridges this gap with its flexibility in shape and concentration and its multiplicative diffusion processes, allowing it to better approximate the true data distributions of various types.
>
> While our GBD model does not explicitly regularize motif similarity, its ability to capture the statistical characteristics of graph elements enhances the topological similarity between real and generated graphs, including motifs. This is reflected in high V.U.N. and Scaf scores observed on Planar, SBM and molecular datasets, which are metrics directly related to motif similarity.
> Additionally, beta diffusion exhibits the "destiny driven" property identified in recent work (GruM), which has been linked to improved graph quality and faster convergence, further demonstrating its advantages in graph generation.

---

> ### Author Response · Authors · 2024-11-21
>
> 4. **Regarding the choice of $\omega$:**
>
> We’d like to clarify that we used the default value of $\omega = 0.01$, as it had already yielded state-of-the-art results. However, following your suggestion, we have conducted an ablation study on Planar datasets. Our findings indicate that while performance remains robust across a wide range of $\omega$ values, there is potential for improvement by fine-tuning this parameter. Specifically, as shown in the table below, performance improves from 92.5% to 95% when adjusting $\omega$ from 0.01 to 0.1. Therefore, we recommend tuning $\omega$ if computational resources permit. If resources are limited, maintaining $\omega$ at 0.01 while focusing on other hyperparameters—such as concentration parameters—that are more closely related to the specific characteristics of the graph data, may be advantageous.
>
> **V.U.N.(%) on Planar**
>
> |   $\omega$  | 0   | 0.01 | 0.1 | 0.5 | 1   |
> |------|-----|------|-----|-----|-----|
> | **GBD** | 80  | 92.5 | 95  | 90  | 85  |
>
> 5. **Regarding predicting $E[G_0|G_t]$ in diffusion models:**
>
> We clarify that predicting $E[G_0|G_t]$  is standard within the EDM framework.
> In Gaussian diffusion, commonly used in the DDPM framework, it is also typical to predict the noise, $\epsilon$, rather than $x_0$. As demonstrated in EDM (Karras et al., Elucidating the design space of diffusion based generative models. NeurIPS, 2022), these approaches under Gaussian diffusion are essentially equivalent, with the primary distinction being the method of network preconditioning. Both $x_0$ prediction and $\epsilon$ prediction are widely employed strategies in Gaussian diffusion. However, $x_0$ prediction is more suitable under beta diffusion. This is because the beta distribution is not reparameterizable, and there is no direct analogy for added noise as there is in Gaussian diffusion.
>
> 6. **Details about concentration modulation.**
>
> We choose $\eta$ based on training data statistics during training and use the same prior statistics to define the $\eta$ for sampling. Please refer to the global response for more details. We expect the nodes of the generated graph to be stochastically ordered based on their corresponding order of $\eta$. Since our model is permutation equivariant and takes only the indirect guidance from $\eta$, the predefined order does not affect the final diversity and quality during sampling. Empirically, our model can generate more high-quality and valid graphs with this technique as shown on various experiments and visualization.

---

> > ### Comment · Reviewer_gBaw · 2024-11-26
> >
> > Thank you for your response, which has clarified several aspects of your work. I have raised the contribution rating and overall rating. I still share the concern with another reviewer that the equations for beta diffusion are not paired with sufficient explanation and intuition, and I believe the presentation could be improved.

---

> > > ### Author Response · Authors · 2024-11-28
> > >
> > > We thank reviewer for raising the initial ratings to the acceptance line and thanks for your again your valuable advice on improving our paper.
> > >
> > > Following your suggestion, we will update the background about beta diffusion in the revision and we look forward to having more conversations with you to improve this work.

---

> ### Comment · Area_Chair_LinD · 2024-11-25
>
> Could please acknowledge and respond to the rebuttal.

---

### Author Response · Authors · 2024-11-21
**Global Response**

# Response to all:

We sincerely thank all reviewers for their thorough reviews and insightful questions. Upon reviewing the feedback, we noticed two shared concerns: (1) Reviewers gBaw and 9sdC highlighted the need for greater clarity regarding the role of concentration modulation, and (2) Reviewer gBaw, and PnYi raised questions about the motivation for using beta diffusion over other diffusion models. Below, we provide a general response to these two shared concerns, followed by detailed responses to each individual reviewer.

**(1) Concentration Modulation**

We first note that Concentration Modulation is a distinctive feature of beta diffusion, which we have innovatively leveraged in this study to inject graph-specific inductive biases. By applying different concentration parameters $\eta$ to various nodes and edges, we effectively inject inductive biases related to node degree or edge connectivity as desired. This novel approach allows us to precisely tailor the diffusion process to the specific structural characteristics of the graph.

In Gaussian/categorical diffusion, controlling the attenuation of the mean and the signal-to-noise ratio (SNR) at any given time $t$ is achieved solely by adjusting the noise schedule $\alpha_t$. In contrast, in beta diffusion, while the attenuation of the mean is still determined by $\alpha_t$, the SNR at time $t$, expressed as
$$
\frac{\alpha_t\cdot G_0\cdot (\eta+1)}{1-\alpha_t\cdot G_0},
$$
is not only influenced by $\alpha_t$ but also directly proportional to $\eta+1$. In addition, it is influenced by $G_0$, which represents the strength of the underlying clean signal.

The concentration parameter $\eta$ adds another degree of freedom, allowing for finer control over the SNR. A higher $\eta$ increases the SNR, enabling the signal to emerge earlier. Thus by assigning different $\eta$ to various nodes and edges, we can strategically encourage the emergence of important structures earlier among the nodes and edges with larger $\eta$ in the diffusion process.

We further examine the role of the Concentration Parameter and the estimated $G_0$ during the reverse process, as described in Equations (6) and (7). The signal sampled at each step of the reverse process has a signal-to-noise ratio (SNR) expressed as:

$$(\alpha_{t-1} - \alpha_t) \frac{1 - \alpha_t \hat{G_0}}{1 - \alpha_{t-1}  \hat{G}_0} \eta \hat{G}_0,$$

where $\hat{G_0} = \hat{G}_{\theta}(G_t, t)$ represents the estimated clean signal based on the sampled noisy signal at time $t$. Consequently, a larger $\eta$ can enhance the retention of a strong signal once it is sampled at a given time step, encouraging its influence to persist through subsequent reverse steps.

Below we provide further clarifications on how concentration modulation is applied for graph generation.
It is worth noting that this technique requires defining $\eta$ during both the training and sampling stages, where the priors are typically derived from summary statistics in the graph training data.  During training, we can easily initialize the concentration parameter for each node and edge as detailed in Section 2.2 and Appendix C.2.

During the sampling stage, we randomly apply summary statistics from the training data and use them to initialize the concentration parameters for each unknown node and edge, as done during the training stage. We note that the cost of applying the summary statistics obtained from the training data during sampling is negligible.
This approach leads to the generation of desirable and diverse graphs, where the summary statistics become correlated with the pre-specified concentration parameters.

For molecule graph generation, we provide a straightforward strategy detailed in Appendix C.2. Utilizing a similar initialization procedure as in general graph generation, we directly define the concentration parameters based on the importance of atom and bond distributions summarized from the training data.

---

> ### Author Response · Authors · 2024-11-21
>
> **(2) The motivation of using beta diffusion**
>
> - Unlike standard diffusion models that rely on Gaussian or categorical distributions, which specialize in either continuous or discrete domains, beta diffusion bridges this gap with its flexibility in shape and concentration, coupled with its multiplicative diffusion processes. This allows it to model a broad spectrum of continuous distributions and effectively approximate discrete distributions as well. By leveraging these unique properties, beta diffusion establishes a robust and adaptable framework ideally suited for the diverse data types encountered in graph generation.
>
> - Graph data elements exhibit diverse probabilistic characteristics: edges in the adjacency matrix can be binary or discrete, while node attributes can be either discrete or continuous, depending on their specific definitions.
> Existing diffusion-based graph generative models built upon Gaussian or categorical distributions, each of which has limitations in handling data with both continuous and discrete elements. Gaussian-based diffusion is well-suited for continuous data but lacks efficiency for discrete structures, while categorical-based diffusion specializes in discrete data but struggles with continuous components.
>
> - Beta diffusion bridges this gap with its flexibility in mean and concentration, coupled with its multiplicative diffusion processes. This allows it to model a broad spectrum of continuous distributions and effectively approximate discrete distributions as well. By leveraging these unique properties, beta diffusion establishes a robust and adaptable framework ideally suited for the diverse data types encountered in graph generation.

---

### Meta-Review · Area_Chair_LinD · 2024-12-19

**Metareview:**

This paper introduces Graph Beta Diffusion (GBD), a generative model designed to handle the unique mix of discrete and continuous components in graph data. By using a beta diffusion process and a modulation technique, GBD generates realistic graphs and outperforms existing models on various benchmarks.

I recommend acceptance of the paper. All reviewers gave a score of 6, and the authors addressed the reviewer's concerns during the rebuttal period.

**Additional Comments On Reviewer Discussion:**

See my comments above.

---

### Decision · Program_Chairs · 2025-01-22

Accept (Poster)